# ACTIVE LEARNING FOR NEURAL PDE SOLVERS

**Daniel Musekamp**[1,3]  **Marimuthu Kalimuthu**[1,2,3]  **David Holzmüller**[4]
**Makoto Takamoto**[5]  **Mathias Niepert**[1,2,3,5]

[1]University of Stuttgart  [2]SimTech  [3] IMPRS-IS
[4]INRIA Paris, Ecole Normale Supérieure, PSL University  [5]NEC Labs Europe
`daniel.musekamp@ki.uni-stuttgart.de`

## ABSTRACT

Solving partial differential equations (PDEs) is a fundamental problem in science and engineering. While neural PDE solvers can be more efficient than established numerical solvers, they often require large amounts of training data that is costly to obtain. Active learning (AL) could help surrogate models reach the same accuracy with smaller training sets by querying classical solvers with more informative initial conditions and PDE parameters. While AL is more common in other domains, it has yet to be studied extensively for neural PDE solvers. To bridge this gap, we introduce AL4PDE, a modular and extensible active learning benchmark. It provides multiple parametric PDEs and state-of-the-art surrogate models for the solver-in-the-loop setting, enabling the evaluation of existing and the development of new AL methods for neural PDE solving. We use the benchmark to evaluate batch active learning algorithms such as uncertainty- and feature-based methods. We show that AL reduces the average error by up to 71% compared to random sampling and significantly reduces worst-case errors. Moreover, AL generates similar datasets across repeated runs, with consistent distributions over the PDE parameters and initial conditions. The acquired datasets are reusable, providing benefits for surrogate models not involved in the data generation.

## 1 INTRODUCTION

Partial differential equations describe numerous physical phenomena such as fluid dynamics, heat flow, and cell growth. Because of the difficulty of obtaining exact solutions for PDEs, it is common to utilize numerical schemes to obtain approximate solutions. However, numerical solvers require a high temporal and spatial resolution to obtain sufficiently accurate numerical solutions, leading to high computational costs. This issue is further exacerbated in settings like parameter studies, inverse problems, or design optimization, where many iterations of simulations must be conducted. Thus, it can be beneficial to replace the numerical simulator with a surrogate model by training a neural network to predict the simulator outputs (Takamoto et al., 2022; Lippe et al., 2023; Brandstetter et al., 2021; Gupta & Brandstetter, 2023; Li et al., 2021b). In addition to being more efficient, neural surrogate models have other advantages, such as being end-to-end differentiable.

One of the main challenges of neural PDE surrogates is that their training data is often obtained from the same expensive simulators they are intended to ultimately replace. Hence, training a surrogate provides a computational advantage only if the generation of the training data set requires fewer simulations than will be saved during inference. Moreover, it is non-trivial to obtain training data for a diverse set of initial conditions and PDE parameters required to train a surrogate with sufficient generalizability. For instance, contrary to training foundation models for text and images, foundation models for solving PDEs require targeted and expensive data generation to generalize well.

Active learning is a possible solution to these challenges as it might help to iteratively select a smaller number of the most informative and diverse training trajectories, thereby reducing the total number of simulations required to reach the same level of accuracy. Furthermore, AL may also improve the reliability of the surrogate models by covering challenging dynamical regimes with enough training data, which may otherwise be hard to find through hand-tuned input distributions. However, generating data for neural PDE solvers is a challenging problem for AL due to the complex regression

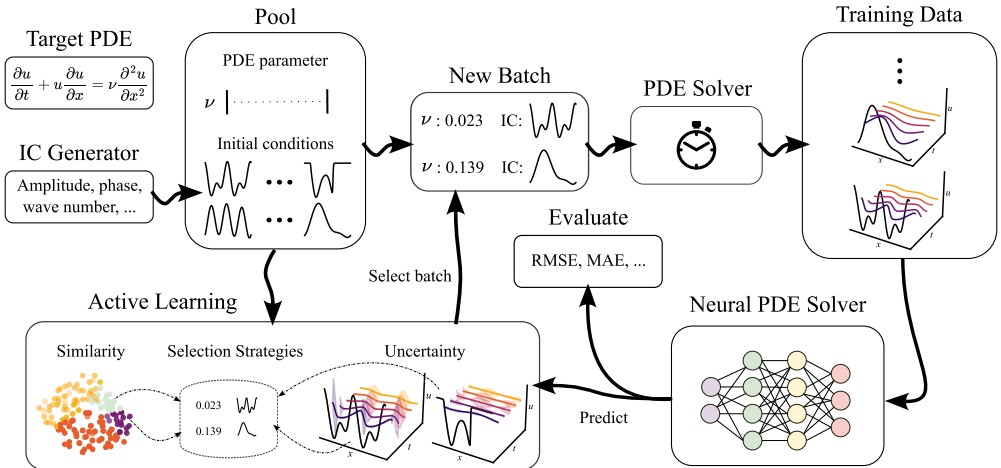

Figure 1: An extensible benchmark framework for pool-based active learning for neural PDE solvers.

tasks characterized by high-dimensional input and output spaces and time series data. While AL has been used extensively for other scientific ML domains such as material science (Lookman et al., 2019; Wang et al., 2022; Zaverkin et al., 2022; 2024), it has only been recently applied to PDEs in the context of physics-informed neural networks (PINNs; Wu et al. 2023a; Sahli Costabal et al. 2020; Aikawa et al. 2023), specific PDE domains (Pestourie et al., 2021; 2023), or direct prediction models (Li et al., 2024a; 2021b). Hence, AL is still unexplored for a broader class of neural PDE solvers, which currently rely on extensive, brute-force numerical simulations to generate a sufficient amount of training data.

**Contributions.** This paper presents AL4PDE, the first AL framework for neural PDE solvers. The benchmark supports the study of existing AL algorithms in scientific ML applications and facilitates the development of novel PDE-specific AL methods. In addition to various AL algorithms, the benchmark provides differentiable numerical simulators for multiple PDEs, such as compressible Navier-Stokes, and neural surrogate models, such as the U-Net (Ronneberger et al., 2015; Gupta & Brandstetter, 2023). The benchmark is extensible, allowing new algorithms, models, and tasks to be added. Using the benchmark, we conducted several experiments exploring the behavior of AL algorithms for PDE solving. These experiments show that AL can increase data efficiency and especially reduce worst-case errors. Among the methods, largest cluster maximum distance (LCMD, Holzmüller et al., 2023), stochastic batch active learning (SBAL, Kirsch et al., 2023) and batch active learning via information matrices (BAIT, Ash et al., 2021) are the best-performing algorithms. We demonstrate that using AL can result in more accurate surrogate models trained in less time. Additionally, the generated data distribution is consistent between random repetitions, initial datasets, and models, showing that AL can reliably generate reusable datasets for neural PDE solvers that were not used to gather the data. The code is available at https://github.com/dmusekamp/al4pde.

## 2 BACKGROUND

We seek the solution $\mathbf{u} : [0,T] \times \mathcal{X} \to \mathbb{R}^{N_c}$ of a PDE with a $D$-dimensional spatial domain $\mathcal{X}$, $\mathbf{x} = [x_1, x_2, \ldots, x_D]^\top \in \mathcal{X}$, temporal domain $t \in [0,T]$, and $N_c$ field variables or channels $c$ (Brandstetter et al., 2021):

$$\partial_t \mathbf{u} = F\left(\boldsymbol{\lambda}, t, \mathbf{x}, \mathbf{u}, \partial_{\mathbf{x}} \mathbf{u}, \partial_{\mathbf{x}\mathbf{x}} \mathbf{u}, \ldots\right), \qquad (t, \mathbf{x}) \in [0,T] \times \mathcal{X} \tag{1}$$

$$\mathbf{u}(0, \mathbf{x}) = \mathbf{u}^0(\mathbf{x}), \quad \mathbf{x} \in \mathcal{X}; \qquad \mathcal{B}[\mathbf{u}](t, \mathbf{x}) = 0, \quad (t, \mathbf{x}) \in [0,T] \times \partial\mathcal{X} \tag{2}$$

Here, the boundary condition $\mathcal{B}$ (Eq. 2) determines the behavior of the solution at the boundaries $\partial\mathcal{X}$ of the spatial domain $\mathcal{X}$, and the initial condition (IC) $\mathbf{u}^0$ defines the initial state of the system (Eq. 2). The vector $\boldsymbol{\lambda} = (\lambda_1, ..., \lambda_l)^\top \in \mathbb{R}^l$ with $\lambda_i \in [a_i, b_i]$ denotes the PDE parameters which influence the dynamics of the physical system governed by the PDE such as the diffusion coefficient in Burgers' equation. The field variables $c$ refer to the different physical quantities modeled in the

solution, e.g. the density and pressure fields in fluid dynamics. In the following, we only consider a single boundary condition for simplicity, and thus, a single initial value problem can be identified by the tuple $\boldsymbol{\psi} = (\mathbf{u}^0, \boldsymbol{\lambda})$. The inputs to the initial value problem are drawn from the test input distribution $p_T$, $\boldsymbol{\psi} \sim p_T(\boldsymbol{\psi}) = p_T(\mathbf{u}^0)p_T(\boldsymbol{\lambda})$. The distributions are typically only given implicitly, i.e., we are given an IC generator $p_T(\mathbf{u}^0)$ and a PDE parameter generator $p_T(\boldsymbol{\lambda})$, from which we can draw samples. For instance, the ICs may be drawn from a superposition of sinusoidal functions with random amplitudes and phases (Takamoto et al., 2022), while the PDE parameters ($\lambda_i$) are typically drawn uniformly from their interval $[a_i, b_i]$.

The ground truth data is generated using a numerical solver, which can be defined as a forward operator $\mathcal{G} : \mathcal{U} \times \mathbb{R}^l \rightarrow \mathcal{U}$, mapping the solution at the current timestep to the one at the next (Li et al., 2021b; Takamoto et al., 2022), $\mathbf{u}(t + \Delta t, \cdot) = \mathcal{G}(\mathbf{u}(t, \cdot), \boldsymbol{\lambda})$ with timestep size $\Delta t$. Here, $\mathcal{U}$ is a suitable space of functions $\mathbf{u}(t, \cdot)$. The solution $\mathbf{u}$ is uniformly discretized across the spatial dimensions, yielding $N_\mathbf{x}$ spatial points in total and the temporal dimension into $N_t$ timesteps. The forward operator is applied autoregressively, i.e., feeding the output state back into $\mathcal{G}$ (also called rollout), to obtain a full trajectory $\mathbf{u} = (\mathbf{u}^0, \mathbf{u}^1, ..., \mathbf{u}^{N_t})$. We aim to replace the numerical solver with a neural PDE solver. While there are also other paradigms such as PINNs (Raissi et al., 2019), we restrict ourselves to autoregressive solvers $\mathcal{G}_\theta$ with $\hat{\mathbf{u}}(t + \Delta t, \cdot) = \mathcal{G}_\theta(\hat{\mathbf{u}}(t, \cdot), \boldsymbol{\lambda})$. The training set for the said $\mathcal{G}_\theta(\mathbf{u}(t, \cdot), \boldsymbol{\lambda})$ consists of aligned pairs of $\boldsymbol{\psi}$ and the corresponding solutions obtained from the numerical solver, i.e., $\mathcal{S}_{\text{train}} = \{(\boldsymbol{\psi}_1, \mathbf{u}_1), \ldots, (\boldsymbol{\psi}_{N_{\text{train}}}, \mathbf{u}_{N_{\text{train}}})\}$. The neural network parameters $\theta$ are minimized using root mean squared error (RMSE) on training samples,

$$\mathcal{L}_{\text{RMSE}}(\mathbf{u}, \hat{\mathbf{u}}) = \sqrt{\frac{1}{N_t N_\mathbf{x} N_c} \sum_{i=1}^{N_t} \sum_{j=1}^{N_\mathbf{x}} \|\mathbf{u}(t_i, \mathbf{x}_j) - \hat{\mathbf{u}}(t_i, \mathbf{x}_j)\|_2^2}, \tag{3}$$

where $\hat{\mathbf{u}}$ denotes the estimated solutions of the neural surrogate models.

## 3 RELATED WORK

Neural surrogate models for solving parametric PDEs are a popular area of research (Takamoto et al., 2023; Kapoor et al., 2023; Lippe et al., 2023; Cho et al., 2024). Most existing works, however, often focus on single or uniformly sampled parameter values for the PDE coefficients and improving the neural architectures to boost the accuracy. In the context of neural PDE solvers, AL has primarily been applied to select the collocation points of PINNs. A typical approach is to sample the collocation points based on the residual error directly (Arthurs & King, 2021; Gao & Wang, 2023; Mao & Meng, 2023; Wu et al., 2023a). While this strategy can be effective, it differs from standard AL since it uses the "label", i.e., the residual loss, when selecting data points. In this line of work, Bruna et al. (2024) use AL to select collocation points for Neural Galerkin Schemes. Aikawa et al. (2023) use a Bayesian PINN to select points based on uncertainty, whereas Sahli Costabal et al. (2020) employ a PINN ensemble for AL of cardiac activation mapping.

Pestourie et al. (2020) use AL to approximate Maxwell equations using ensemble-based uncertainty quantification for metamaterial design. Uncertainty-based AL was also employed for diffusion, reaction-diffusion, and electromagnetic scattering (Pestourie et al., 2023). In multi-fidelity AL, the optimal spatial resolution of the simulation is chosen (Li et al., 2020; 2021a; Wu et al., 2023b). For instance, Li et al. (2024a) use an ensemble of FNOs in the single prediction setting. Wu et al. (2023c) apply AL to stochastic simulations using a spatio-temporal neural process. Bajracharya et al. (2024) investigate AL to predict the stationary solution of a diffusion problem. They consider AL using two different uncertainty estimation techniques and selecting based on the diversity in the input space. Pickering et al. (2022) use AL to find extreme events using ensembles of DeepONets (Lu et al., 2021). Gajjar et al. (2022) provide theoretical results for AL of PDEs for single neuron models. Closely related to AL is the field of design of experiments (DoE, Garud et al., 2017; Qu, 2023; Huan et al., 2024) which has also been applied to neural PDE solvers (Wu et al., 2023a; Li et al., 2024b). For example, space-filling, static DoE methods such as Latin Hypercube sampling (McKay et al., 1979) can be applied to avoid clustering induced by pure random sampling of the PDE input parameters (Wu et al., 2023a; Li et al., 2024b; Chandra et al., 2024). Next to using AL, it is also possible to reduce the data generation time using Krylov subspace recycling (Wang et al., 2023) or by applying data augmentation techniques such as Lie-point symmetries (Brandstetter et al., 2022). Such symmetries could also be combined with AL using LADA (Kim et al., 2021).

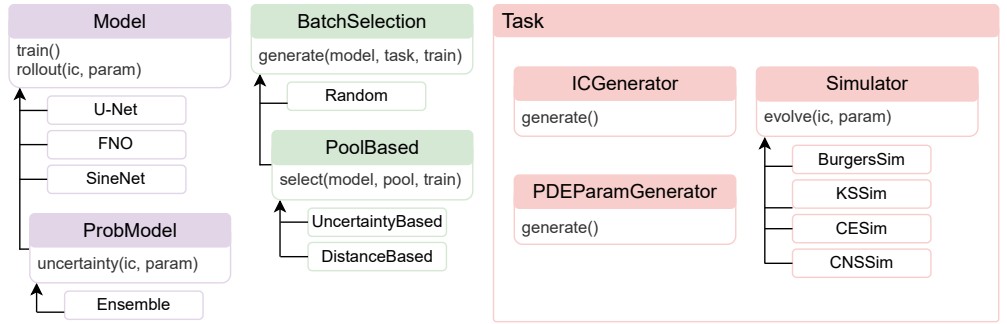

Figure 2: Structural overview of the AL4PDE benchmark.

In recent years, several benchmarks for neural PDE solvers have been published (Takamoto et al., 2022; Gupta & Brandstetter, 2023; Zhongkai et al., 2025; Luo et al., 2023; Liu et al., 2024). For instance, PDEBench (Takamoto et al., 2022) and PDEArena (Gupta & Brandstetter, 2023) provide efficient implementations of numerical solvers for multiple hydrodynamic PDEs such as Advection, Navier-Stokes, as well as recent implementations of neural PDE solvers (e.g., DilResNet, U-Net, FNO) for standard and conditioned PDE solving. Similarly, CODBench (Burark et al., 2024) compares the performance of different neural operators. WaveBench (Liu et al., 2024) is a benchmark specifically aimed at wave propagation PDEs that are categorized into time-harmonic and time-varying problems. For a more detailed discussion of related benchmarks, see Appendix A.3. Contrary to prior work, AL4PDE is the first framework for evaluating and developing AL methods for neural PDE solvers.

# 4 AL4PDE: AN AL FRAMEWORK FOR NEURAL PDE SOLVERS

The AL4PDE benchmark consists of three major parts: (1) AL algorithms, (2) surrogate models, and (3) PDEs and the corresponding simulators. It follows a modular design to make the addition of new approaches or problems as easy as possible (Fig. 2). The following sections describe the AL approaches, including the general problem setup, acquisition and similarity functions, and batch selection strategies. Moreover, we describe the included PDEs and surrogate models.

## 4.1 PROBLEM DEFINITION AND SETUP

AL aims to select the most informative training samples so that the model can reach the same generalization error with fewer calls to the numerical solver. We measure the error using test trajectories on random samples from an input distribution $p_T$. Fig. 1 shows the full AL cycle. Since it requires retraining the NN(s) after each round, we use batch AL with sufficiently large batches. Specifically, in each round, a batch of simulator inputs $S_{\text{batch}} = \{\psi_1, ..., \psi_{N_{\text{batch}}}\}$ is selected. It is then passed to the numerical solver, which computes the output trajectories using numerical approximation schemes. The new trajectories are then added to the training set, and the cycle is repeated.

We implement *pool-based* active learning methods, which select from a set of possible inputs $S_{\text{pool}} = \{\psi_1, ..., \psi_{N_{\text{pool}}}\}$ called "pool". The selected batch is then removed from the pool, simulated, and added to the training set $S_{\text{train}}$:

$$S_{\text{pool}} \leftarrow S_{\text{pool}} \setminus S_{\text{batch}}, \qquad S_{\text{train}} \leftarrow S_{\text{train}} \cup \text{solve}(S_{\text{batch}}). \qquad (4)$$

We sample the pool set randomly from a proposal distribution $\pi$. In our experiments, we sample pool and test set from the same input distribution $\pi = p_T$, although $p_T$ might not always be known in practice. Following common practice, the initial batch is selected randomly. Besides pool-based methods, our framework is also compatible with query-synthesis AL methods that are not restricted to a finite pool set. Several principles are useful for the design of AL methods (Wu, 2018): First, they should select highly *informative* samples that allow the model to reduce its uncertainty. Second, selecting inputs that are *representative* of the test input distribution at test time is often desirable. Third, the batch should be *diverse*, i.e., the individual samples should provide non-redundant information. The last point is particular to the batch setting, which is essential to maintain acceptable runtimes. In the following, we will investigate batch AL methods that first extract latent features or direct

uncertainty estimates from the neural surrogate model for each sample in the pool and subsequently apply a selection method to construct the batch.

## 4.2 UNCERTAINTIES AND FEATURES

Since neural PDE solvers provide high-dimensional autoregressive rollouts without direct uncertainty predictions, many AL methods cannot be applied straightforwardly. In the AL4PDE framework, we select the following two different classes of methods: the uncertainty-based approach, which directly assigns an uncertainty score to each candidate, and the feature-based framework of Holzmüller et al. (2023), which uses features (or kernels) to evaluate the similarity between inputs.

**Uncertainties.** Epistemic uncertainty is often used as a measure of sample informativeness. While a more costly Bayesian approach is possible, we adopt the query-by-committee (QbC) approach (Seung et al., 1992), a simple but effective method that utilizes the variance between the ensemble members' outputs as an uncertainty estimate:

$$a_{\text{QbC}}(\boldsymbol{\psi}_i) := \frac{1}{N_t N_{\mathbf{x}} N_c} \sum_{j=1}^{N_t} \sum_{k=1}^{N_{\mathbf{x}}} \frac{1}{N_m} \sum_{m=1}^{N_m} \|\hat{\mathbf{u}}_{i,m}(t_j, \mathbf{x}_k) - \overline{\hat{\mathbf{u}}_i}(t_j, \mathbf{x}_k)\|_2^2 . \tag{5}$$

Here, $\overline{\hat{\mathbf{u}}_i}$ is the mean prediction of all $N_m$ models with $\overline{\hat{\mathbf{u}}_i}(t, \mathbf{x}) = \sum_m \hat{\mathbf{u}}_{i,m}(t, \mathbf{x})/N_m$. The ensemble members produce different outputs $\hat{\mathbf{u}}_i$ due to the inherent randomness resulting from the weight initialization and stochastic gradient descent. The assumption of QbC is that the variance of the ensemble member predictions correlates positively with the error. A high variance, therefore, points to a region of the input space where we need more data. Using the variance of the model outputs directly corresponds to minimizing the expected MSE. Note that many more error metrics can be considered for PDEs (Takamoto et al., 2022), for which measures other than the variance may be more appropriate.

**Features.** Many deep batch AL methods rely on some feature representation $\phi(\boldsymbol{\psi}) \in \mathbb{R}^p$ of inputs and utilize a distance metric in the feature space as a proxy for the similarity between inputs, which can help to ensure diversity of the selected batch. A typical representation is the inputs to the last neural network layer, but other representations are possible (Holzmüller et al., 2023). For neural PDE solvers, we compute the trajectory and concatenate the last-layer features at each timestep. Since this can result in very high-dimensional feature vectors, we follow Holzmüller et al. (2023) and apply Gaussian sketching. Specifically, we use $\phi_{\text{sketch}}(\boldsymbol{\psi}) := \mathbf{U}\phi(\boldsymbol{\psi})/\sqrt{p'} \in \mathbb{R}^{p'}$, to reduce the feature space to a fixed dimension $p'$ using a random matrix $\mathbf{U} \in \mathbb{R}^{p' \times p}$ with i.i.d. standard Gaussian entries.

While ensemble-based AL methods can also be formulated in terms of feature maps (Kirsch, 2023), the use of latent features allows AL methods to work with a single model. Moreover, methods based on distances of latent features can naturally incorporate diversity into batch AL by avoiding the selection of highly similar examples. Feature-based AL methods are, however, not translation-invariant. In the considered settings with periodic boundary conditions, an IC translated along the spatial axis will produce a trajectory shifted by the same amount. By using periodic padding within the convolutional layers of U-Net, the network is equivariant w.r.t. translations; hence, adding a translated version of the same IC is redundant. Uncertainty-based approaches based on ensembles are translation-invariant since all ensemble model outputs are shifted by the same amount and produce the same outputs. To make feature-based AL translation invariant, we take the spatial average over the features.

## 4.3 BATCH SELECTION STRATEGIES

Given uncertainties or features, we need to define a method to select a batch of pool samples. As a generic baseline, we compare to the selection of a (uniformly) **random** sampling of the inputs according to the input distribution, $\boldsymbol{\psi} \sim p_T(\boldsymbol{\psi})$. Additionally, we include Latin Hypercube Sampling (**LHS**) as a static DoE baseline (McKay et al., 1979).

**Uncertainty-based selection strategies.** When given a single-sample acquisition function $a$ such as the ensemble uncertainty, a simple and common approach to selecting a batch of $k$ samples is

**Top-K**, taking the $k$ most uncertain samples. However, this does not ensure that the selected batch is diverse. To improve diversity, Kirsch et al. (2023) proposed stochastic batch active learning (**SBAL**). SBAL samples inputs $\psi$ from the remaining pool set $\mathcal{S}_{\text{pool}}$ without replacement according to the probability distribution $p_{\text{power}}(\psi) \propto a(\psi)^m$, where $m$ is a hyperparameter controlling the sharpness of the distribution. Random sampling corresponds to $m = 0$ and Top-K to $m = \infty$. The advantage of SBAL is that it selects samples from input regions that are not from the highest mode of the uncertainty distribution and encourages diversity.

**Feature-based selection strategies.** In the simpler version of their **Core-Set** algorithm, Sener & Savarese (2018) iteratively select the input from the remaining pool with the highest distance to the closest selected or labeled point. While Core-Set produces batches of diverse and informative samples, its objective is to cover the feature space uniformly. Hence, Core-Set, in general, does not select samples that are representative of the proposal distribution. To alleviate this issue, Holzmüller et al. (2023) propose to replace the greedy Core-Set with **LCMD**, a similarly efficient method inspired by k-medoids clustering. LCMD interprets previously selected inputs as cluster centers, assigns all remaining pool points to their closest center, selects the cluster with the largest sum of squared distances to the center, and from this cluster selects the point that is furthest away from the center. The newly selected point then becomes a new center and the process is repeated until a batch of the desired size is obtained. While finding efficient Bayesian AL methods for our setting with high-dimensional outputs and autoregressive generation is challenging, we can apply efficient Bayesian AL methods to the proxy task of single-output Bayesian linear regression on given features. In particular, **BAIT** (Ash et al., 2021) aims to minimize the average posterior predictive variance (Holzmüller et al., 2023). We apply BAIT to the same aggregated features as LCMD and Core-Set.

## 4.4 PDEs

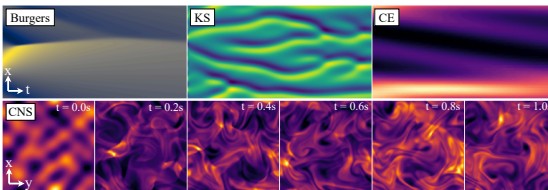

| PDE | T in s | Sim. Res. $(N_t, N_x, [N_y], [N_z])$ | Train. Res. $(N_t, N_x, [N_y], [N_z])$ |
|---|---|---|---|
| Burgers | 2 | (201, 1024) | (41, 256) |
| KS | 40 | (801, 512) | (41, 256) |
| CE | 4 | (501, 64) | (51, 64) |
| 1D CNS | 2 | (201, 512) | (41, 128) |
| 2D CNS | 1 | (21, 128, 128) | (21, 64, 64) |
| 3D CNS | 1 | (21, 64, 64, 64) | (21, 32, 32, 32) |

Figure 3: Example trajectories of the PDEs.  Table 1: Discretizations of the PDEs.

We consider 1D, 2D, and 3D parametric PDEs with periodic boundary conditions except for 1D CNS. Table 1 lists the selected resolutions. The first 1D PDE is the **Burgers'** equation from PDEBench (Takamoto et al., 2022) with kinematic viscosity $\nu$: $\partial_t u + u \partial_x u = (\nu/\pi)\partial_{xx}u$. Secondly, the Kuramoto–Sivashinsky (**KS**) equation, $\partial_t u + u \partial_x u + \partial_{xx}u + \nu\partial_{xxxx}u = 0$, from Lippe et al. (2023) demonstrates diverse dynamical behaviors, from fixed points and periodic limit cycles to chaos (Hyman & Nicolaenko, 1986). Next to the viscosity $\nu$, the domain length L is also varied. Thirdly, to test a multiphysics problem with more parameters, we include the so-called combined equation (**CE**) from Brandstetter et al. (2021) where we set the forcing term $\delta = 0$: $\partial_t u + \partial_x\left(\alpha u^2 - \beta\partial_x u + \gamma\partial_{xx}u\right) = 0$. Depending on the value of the PDE coefficients $(\alpha, \beta, \gamma)$, this equation recovers the Heat, Burgers, or the Korteweg-de-Vries PDE. Finally, we use the compressible Navier-Stokes (**CNS**) equations in 1D, 2D, and 3D from PDEBench (Takamoto et al., 2022), $\partial_t\rho + \nabla \cdot (\rho\mathbf{v}) = 0$, $\rho(\partial_t\mathbf{v} + \mathbf{v} \cdot \nabla\mathbf{v}) = -\nabla p + \eta\triangle\mathbf{v} + (\zeta + \eta/3)\nabla(\nabla \cdot \mathbf{v})$, $\partial_t(\epsilon + \rho v^2/2) + \nabla \cdot [(p + \epsilon + \rho v^2/2)\mathbf{v} - \mathbf{v} \cdot \sigma'] = \mathbf{0}$. The ICs are generated from random initial fields. For 1D CNS, we consider an out-going boundary condition. Full details on PDEs, ICs, and the PDE parameter distributions can be found in Appendix B.

## 4.5 Neural Surrogate Models

Currently, the benchmark includes the following neural PDE solvers: (i) a recent version of U-Net (Ronneberger et al., 2015) from Gupta & Brandstetter (2023), (ii) SineNet (Zhang et al., 2024), which is an enhancement of the U-Net model that corrects the feature misalignment issue in the

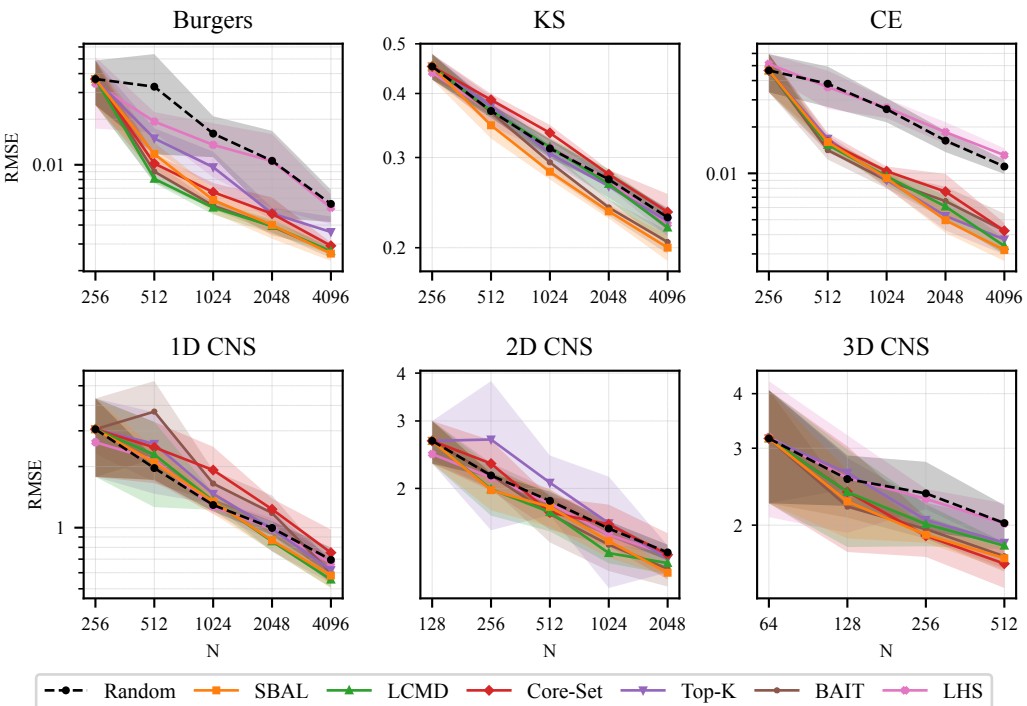

Figure 4: Error over the number of trajectories in the training set (N). The shaded area represents the $95\%$ confidence interval of the mean calculated over multiple seeds. AL can reduce the error relative to random sampling of the inputs on all tested PDEs but CNS, where the difference was not significant.

residual connections of modern U-Nets and can be considered a model with state-of-the-art accuracy, specifically for advection-type equations, and (iii) the Fourier neural operator (FNO, Li et al., 2021b).

## 5    SELECTION OF EXPERIMENTS

We investigate (i) the impact of AL methods on the average error, (ii) the error distribution, (iii) the variance and reusability of the generated data, (iv) the temporal advantage of AL, and (v) conduct an ablation study concerning the different design choices of SBAL and LCMD. We use a smaller version of the modern U-Net from Gupta & Brandstetter (2023). We train the model on sub-trajectories (two steps) to strike a balance between learning auto-regressive rollouts and fast training. For 1D CNS, we found a trajectory length of four to be necessary for stable training. The training is performed for 500 epochs with a cosine schedule, which reduces the learning rate from $10^{-3}$ to $10^{-5}$. The batch size is set to 512 (2D CNS: 64). We use an exponential data schedule, i.e., in each AL iteration, the amount of data added is equal to the current training set size (Kirsch et al., 2023). For 1D equations, we start with 256 trajectories. The pool size is fixed to 100,000 candidates (3D: 30000). The uncertainty is estimated using two ensemble members (for a fair comparison, just the first model of the ensemble is used to measure the error). For Burgers, we choose the parameter space $\nu \in [0.001, 1)$ and sample values uniformly at random but on a logarithmic scale. For the KS equation, besides the viscosity $\nu \in [0.5, 4)$, we vary the domain length $L \in [0.1, 100)$ as the second parameter. For CE, the parameter space is defined to be $\alpha \in [0, 3)$, $\beta \in [0, 0.4)$, $\gamma \in [0, 1)$. For the 1D and 2D CNS equations, we set $\eta, \zeta \in [10^{-4}, 10^{-1})$ and draw values on a logarithmic scale as with the Burgers' PDE. Additionally, we use random Mach numbers $m \in [0.1, 1)$ for the IC generator. We repeat the experiments with five random seeds (Burgers: ten, 3D CNS: 3) and report the $95\%$ confidence interval of the mean unless stated otherwise. The test set consists of 2048 trajectories simulated with random inputs drawn from $p_T(\psi)$. Due to the memory and compute-intensive nature of 3D

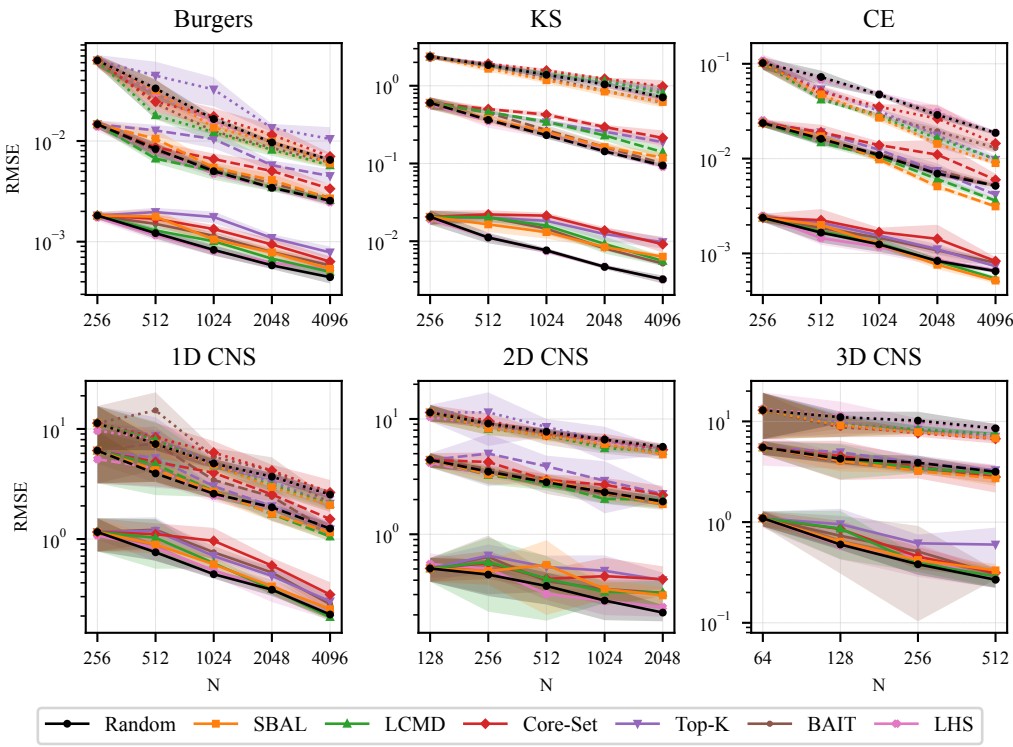

Figure 5: Error quantiles over the number of trajectories in the training set (N). The 50%, 95%, and 99% quantiles are displayed using full, dashed, and dotted lines, respectively. AL especially improves the higher error quantiles, making the trained model more reliable.

time-dependent PDEs, we had to use smaller train and test sets, as well as choose different model and training parameters and use a conditional version of the 3D FNO model (see Appendix C).

**Comparison of AL methods.** Figure 4 shows the RMSE for the various AL methods and PDEs. AL often reduces the error compared to sampling uniformly at random for the same amount of data. The advantage of AL is especially large for CE, which is likely due to the diverse dynamic regimes found in the PDE. SBAL, BAIT, and LCMD achieve similar errors on all PDEs with the exception of KS, where only SBAL and BAIT can improve over random sampling. AL can reach lower error values with only a quarter of the data points in the case of CE and Burgers. However, the greedy methods Top-K and Core-Set even increase the error for some PDEs. We did not find a notable difference between the static DoE method LHS and random sampling. The difference in the CNS tasks was not significant, likely due to the performance of the base model training (see Fig. 8a) for a stronger model. Worst-case errors are of special interest when solving PDEs. Since we found the absolute maximum error to be unstable, we show the RMSE quantiles in Figure 5. Notably, all AL algorithms reduce the higher quantiles while the 50% percentile error is increased in some cases.

**Different Error Functions.** It is important to consider error metrics for surrogate model training besides the RMSE (Takamoto et al., 2022). Thus, we explore the impact of AL on the mean absolute error (MAE) as an example of an alternative metric. As depicted in Figure 7a, SBAL, when using the absolute difference between the models as the uncertainty, can also successfully reduce the MAE. However, the MAE does not improve greatly relative to random sampling when the standard variance between the models is used. Hence, it is crucial to tailor the AL method to the relevant metric.

**Generated Datasets.** The marginal distributions of the PDE and the IC generator parameters implicitly sampled from by AL are shown in Figure 6 for CE. These distributions are highly similar for different random seeds, and thus, AL reliably selects similar training datasets. The various AL methods generally sample similar parameter values but can differ substantially in certain regions of

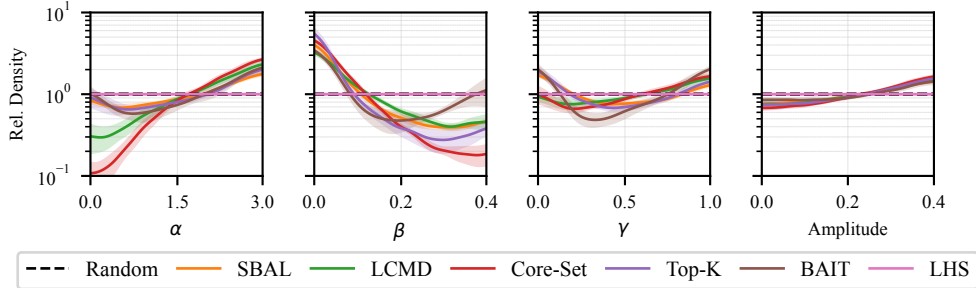

Figure 6: Marginal distribution of the PDE parameters $(\alpha, \beta, \gamma)$ for CE and the amplitudes of the IC in the training set generated by AL for CE (relative to the uniform distribution). The shaded area represents the standard deviation between the random seeds. All AL methods exhibit a small standard deviation, indicating that they reliably generate similar datasets between independent runs.

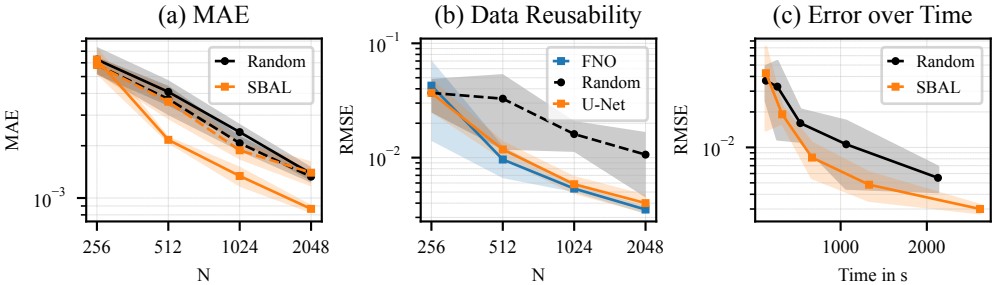

Figure 7: (a) AL with the MAE as the objective on Burgers, compared to the MAE of the same setup trained with the RMSE (dashed). Considering the desired error metric in the uncertainty estimate and training loss is essential. (b) Error of the standard U-Net on Burgers, with data generated using FNO or U-Net with SBAL. The selected data is also helpful for a model not used during AL. (c) Error of the standard U-Net on Burgers over the required total time. Using smaller FNOs to select the data, SBAL can provide smaller errors in the same amount of time.

the parameter space (Appendix G.3). In general, the methods appear to sample more in the region of the chaotic KdV equation ($\alpha = 3, \beta = 0, \gamma = 1$). Appendix G provides the distributions for all PDEs and visual examples. To investigate the effect of the generated data on other models, we use an FNO ensemble to select the data that we use to train the standard U-Net. Figure 7b depicts the error of the U-Net over the number of samples selected using the FNO ensemble, showing the selected data is beneficial for models not used for the AL-based data selection. The reusability of the data is especially important since, otherwise, the whole AL procedure would have to be repeated every time a new model is developed.

**Temporal Behavior.** The main experiments only provide the error over the number of data points since we use problems with rather fast solvers to accelerate the benchmarking of the AL methods. Additionally, a more lightweight model, trained for a shorter time, might be enough for data selection even if it does not reach the best possible accuracy. To investigate AL in terms of time efficiency gains, we perform one experiment on the Burgers' PDE, for which the numerical solver is the most expensive among all 1D PDEs due to its higher resolution. We use SBAL with an ensemble of smaller FNOs (See Appendix C.6 for more details). We train a regular U-Net on the AL collected data, which allows us to use a small, lightweight model for data selection only and an expensive one to evaluate the data selected. Figure 7c shows the accuracy of the evaluation U-Net over the cumulative time consumed for training the selection model, selecting the inputs, and simulation. For the random baseline, only the simulation time is considered. On Burgers, AL provides better accuracy for the same time budget.

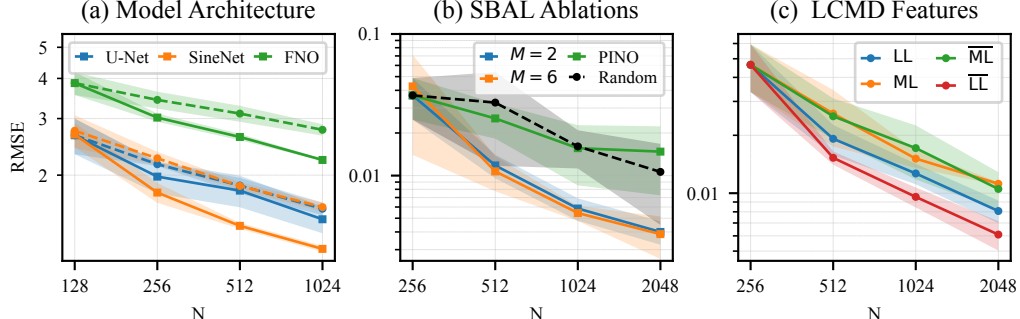

Figure 8: (a) Different base models on 2D CNS using SBAL (solid) and random sampling (dashed). SBAL can also improve the accuracy of other models besides the U-Net. (b) Ablation study on SBAL (Burgers' equation). SBAL already works reliably with only $M = 2$ models in the ensemble. Using the PINO loss (Li et al., 2024c) instead of the ensemble uncertainty does not provide a meaningful uncertainty, as shown by the error on par with random sampling. (c) Comparison of different feature vectors for LCMD on CE. Shown are the last layer feature map ($\underline{LL}$), its spatial average ($\overline{LL}$), as well as the features of the mid layer (ML) and its spatial average ($\overline{ML}$). Averaging the feature maps improves the error, indicating the importance of considering the model invariances.

**Ablations.**  We ablate different design choices for the considered AL algorithms. For the SBAL algorithm, we investigate the base model architecture (Fig. 8a) and the ensemble size (Fig. 8b). On 2D CNS, the accuracy of both SineNet and FNO can be significantly improved using SBAL, showing that AL is also helpful for other architectures. The improvement is even clearer than with the U-Net, which did not show a statistically significant advantage. Consistent with prior work (Pickering et al., 2022), choosing an ensemble size of two models is already sufficient (Fig. 8b). In general, the average uncertainty and error of a trajectory with two ensemble members are correlated with a Pearson coefficient of 0.41 on CE in the worst case up to 0.94 on 2D CNS (Table 10). Adaptive sampling methods utilized in the field of PINNs (Gao & Wang, 2023; Wu et al., 2023a), select collocation points based on the PDE loss. While this is not directly transferable to our setting (Section 3), we try to use the PINO loss (Li et al., 2024c) as an uncertainty estimate in combination with SBAL. As shown in Figure 8b, this is not an effective selection criterion for autoregressive neural PDE solvers. Figure 8c) compares different feature choices for the LCMD algorithm, which are used to calculate the distances. Using the spatial average of the last layer features produces higher accuracy than using the full feature vector or the features from the bottleneck step in the middle of the U-Net. Thus, it is indeed important for distance-based selection to consider the equivariances of the problem in the distance function.

## 6 CONCLUSION

This paper introduces AL4PDE, an extensible framework to develop and evaluate AL algorithms for neural PDE solvers. AL4PDE includes a diverse set of PDEs in 1D, 2D, and 3D spatial dimensions, surrogate models including U-Net, FNO, and SineNet, and AL algorithms such as SBAL and LCMD. An initial study shows that existing AL algorithms can already be advantageous for neural PDE solvers and can allow a model to reach the same accuracy with up to four times fewer data points. Thus, our work shows the potential of AL for making neural PDE solvers more data-efficient and reliable for future application cases. However, the experiments also showed that stable model training can be difficult depending on the base architecture (2D CNS). Such issues especially impact AL since the model is trained repeatedly with different data sets, and the data selection relies on the model. Hence, more work on the reliability of the surrogate model training is necessary. Another general open issue of AL is the question of how to select hyperparameters that work sufficiently well on the growing, unseen datasets during AL. To be closer to realistic engineering applications, future work should also consider more complex geometries and boundary conditions, as well as irregular grids. AL could be especially helpful in such settings due to the inherently more complex input space from which to select.

## REPRODUCIBILITY STATEMENT

The code is available at https://github.com/dmusekamp/al4pde. The repository contains the full configuration files of all reported experiments. Appendix B and C describe the main experimental as well as the model details. For reliable results, we repeat all experiments with ten seeds (Burgers), five seeds (KS, CE, and CNS), and three seeds (3D CNS) and report the 95% confidence interval of the mean unless stated otherwise.

## ACKNOWLEDGEMENTS

We thank the anonymous reviewers on OpenReview whose questions were helpful in improving the manuscript. We acknowledge the support of the German Federal Ministry of Education and Research (BMBF) as part of InnoPhase (funding code: 02NUK078). Marimuthu Kalimuthu and Mathias Niepert are funded by Deutsche Forschungsgemeinschaft (DFG, German Research Foundation) under Germany's Excellence Strategy - EXC 2075 – 390740016. We acknowledge the support of the Stuttgart Center for Simulation Science (SimTech). The authors thank the International Max Planck Research School for Intelligent Systems (IMPRS-IS) for supporting Daniel Musekamp, Marimuthu Kalimuthu, and Mathias Niepert. Moreover, the authors gratefully acknowledge the computing time provided to them at the NHR Center NHR4CES at RWTH Aachen University (project number p0021158). This is funded by the Federal Ministry of Education and Research, and the state governments participating on the basis of the resolutions of the GWK for national high-performance computing at universities (http://www.nhr-verein.de/unsere-partner).

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

# ACTIVE LEARNING FOR NEURAL PDE SOLVERS
## SUPPLEMENTAL MATERIAL

## A  ADDITIONAL BACKGROUND ON RELATED WORK

In this section, we elaborate on related works that tackle active learning in relevant settings and problems discussed here. Moreover, we summarize related work on uncertainty quantification and SciML benchmarks closely related to the proposed AL4PDE benchmark.

### A.1  GENERAL ACTIVE LEARNING

Most AL algorithms are evaluated on classic image classification datasets (Ash et al., 2021; 2019) and many benchmarks also consider the more common classification setting (Rauch et al., 2023; Yang & Loog, 2018; Zhan et al., 2021). There is also work on specialized tasks such as entity matching (Meduri et al., 2020), structural integrity (Moustapha et al., 2022), material science (Wang et al., 2022), or drug discovery (Mehrjou et al., 2021). Holzmüller et al. (2023) present a benchmark for AL of single-output, tabular regression tasks. Wu et al. (2023a) study different adaptive and non-adaptive methods for selecting collocation points for PINNs. Ren et al. (2023) benchmark pool-based AL methods on simulated, mostly tabular regression tasks.

Related to AL is the field of design of experiments  (DoE, Garud et al., 2017; Qu, 2023; Huan et al., 2024). In static DoE, a set of inputs is selected without using feedback from the investigated process. Space-filling DoE methods try to cover the input space optimally without considering any model assumption or feedback (Huan et al., 2024). A pure random sampling of the input space may lead to an undesirable clustering of samples, leading to redundant information. For example, Latin Hypercube sampling (McKay et al., 1979) divides each input variable into equally-spaced intervals, takes one sample from the interval, and then mixes the samples from the different variables. In D-optimal experimental design, the next sample is selected such that the uncertainty in the parameters of a linear regression model is reduced by minimizing the determinant of its Fisher information matrix (Wald, 1943; Kiefer, 1958; Huan et al., 2024). Most similar to AL is sequential DoE. For instance, in sequential Bayesian optimal design, the prior of a Bayesian linear regression model is updated iteratively to the posterior over the model parameters after including the new measurements (Huan et al., 2024). Based on the posterior, a criterion such as the expected information gain (Lindley, 1956; Huan et al., 2024) can be utilized to select the optimal next sample. While such methods have a strong theoretical underpinning, they have been developed for linear regression models and, hence, are not directly applicable to neural networks.

In terms of deep active learning methods for regression, there are multiple approaches: Query-by-committee (Seung et al., 1992) uses ensemble prediction variances as uncertainties. Tsymbalov et al. (2018) use Monte Carlo dropout to obtain uncertainties; however, their method is only applicable by training with dropout. Approaches based on last-layer Bayesian linear regression (Pinsler et al., 2019; Ash et al., 2021) are often convenient since they do not require ensembles or dropout. These methods are applicable in principle in our setting but lose their original Bayesian interpretation since the last layer of a neural operator is applied multiple times during the autoregressive rollout. Distance-based methods like Core-Set (Sener & Savarese, 2018; Geifman & El-Yaniv, 2017) and the clustering-based LCMD (Holzmüller et al., 2023) exhibit better runtime complexity than last-layer Bayesian methods while sharing their other advantages (Holzmüller et al., 2023). Since these algorithms just require some distance function between two input points, we can adapt them to the neural PDE solver setting in Section 4.2.

### A.2  UNCERTAINTY QUANTIFICATION (UQ)

Uncertainty quantification has been studied in the context of SciML simulations. Psaros et al. (2023) provide a detailed overview of UQ methods in SciML, specifically for PINNs and DeepONets. However, effective and reliable UQ methods for neural operators (i.e., mapping between function spaces) and high dimensionality of data, which is common in PDE solving, remain challenging.

**Neural PDE Solvers.**  LE-PDE-UQ (Wu et al., 2024) deals with a method to estimate the uncertainty of neural operators by modeling the dynamics in the latent space. The model has been shown to outperform other UQ approaches, such as Bayes layer, Dropout, and L2 regularization on Navier-Stokes turbulent flow prediction tasks. Unlike the considered setting in our case, the model utilizes

a history of 10 timesteps and has been tested only on a fixed PDE parameter. Hence, it is unclear whether the robustness of this approach remains when these settings change.

Mouli et al. (2024) aim to develop a cost-efficient method for uncertainty quantification of parametric PDEs, specifically one that works well in the out-of-domain test settings of PDE parameters. First, the study shows the challenges of existing UQ methods, such as the Bayesian neural operator (BayesianNO) for out-of-domain test data. It then shows that ensembling several neural operators is an effective strategy for UQ that is well-correlated with prediction errors and proposes diverse neural operators (DiverseNO) as a cost-effective way to estimate uncertainty with just a single model based on FNO outputting multiple predictions.

Thakur (2024) studies UQ in the context of neural operators and develops a probabilistic FNO model to quantify aleatoric and epistemic uncertainties. Weber et al. (2024) study UQ for FNO and propose a Laplace approximation for the Fourier layer to effectively compute uncertainty.

### A.3 Further Scientific Machine Learning Benchmarks

In recent years, various benchmarks and datasets for SciML have been published. We outline some of the major open-source benchmarks below.

PDEBench (Takamoto et al., 2022) is a large-scale SciML benchmark of 1D to 3D PDE equations modeling hydrodynamics ranging from Burgers' to compressible and incompressible Navier-Stokes equations. PDEArena (Gupta & Brandstetter, 2023) is a modern surrogate modeling benchmark including PDEs such as incompressible Navier-Stokes, Shallow Water, and Maxwell equations (Brandstetter et al., 2023). CFDBench (Luo et al., 2023) is a recent benchmark comprising four flow problems, each with three different *operating parameters*, the specific instantiations of which include varying boundary conditions, physical properties, and geometry of the fluid. The benchmark compares the generalization capabilities of a range of neural operators and autoregressive models for each of the said *operating parameters*. LagrangeBench (Toshev et al., 2023) is a large-scale benchmark suite for modeling 2D and 3D fluid mechanics problems based on the Lagrangian specification of the flow field. The benchmark provides both datasets and baseline models. For the former, it introduces seven datasets of varying Reynolds numbers by solving a weak form of NS equations using smoothed particle hydrodynamics. For the latter, efficient JAX implementations of GNN baseline models such as Graph Network-based Simulator and (Steerable) Equivariant GNN are included. EAGLE (Janny et al., 2023) introduces an industrial-grade dataset of non-steady fluid mechanics simulations encompassing 600 geometries and 1.1 million 2D meshes. In addition, to effectively process a dataset of this scale, the benchmark proposes an efficient multi-scale attention model, mesh transformer, to capture long-range dependencies in the simulation. BubbleML (Hassan et al., 2023) is a thermal simulations dataset comprising boiling scenarios that exhibit multiphase and multiphysics phase change phenomena. It also consists of a benchmark validating the dataset against U-Nets and several variants of FNO.

## B Additional Problem Details

In the following section, we will discuss the tasks considered in detail. Table 1 shows the temporal and spatial resolution of the considered PDEs.

### B.1 Burgers' Equation

The 1D Burgers' equation is written as

$$\partial_t u + u \partial_x u = (\nu/\pi)\partial_{xx} u. \tag{6}$$

The spatial domain is set to $x \in [0, 1]$. Following the parameter spacing of the PDE parameters values in PDEBench (Takamoto et al., 2022) and CAPE (Takamoto et al., 2023), we draw them on a logarithmic scale, i.e., we first draw $\lambda_{i,\text{normed}}$ uniformly from $[0, 1)$ and then transform the parameter to its domain $[a_i, b_i)$ using

$$\lambda_i = a_i \exp(\log(b_i/a_i)\lambda_{i,\text{normed}}). \tag{7}$$

We use the FDM-based JAX simulator and the initial condition generator from PDEBench (Takamoto et al., 2022). The ICs are constructed based on a superposition of sinusoidal waves (Takamoto et al.,

2022),

$$\mathbf{u}^0(x) = \sum_{i=1}^{N_w} A_i \sin(2\pi k_i x/L + \phi_i), \tag{8}$$

where the wave number $k_i$ is an integer sampled uniformly from $[1, 5)$, amplitude $A_i$ is sampled uniformly from $[0, 1)$, and phase $\phi_i$ from $[0, 2\pi)$. The number of waves $N_w$ is set to 2. Windowing is applied afterward with a probability of 10%, where all parts of the IC are set to zero outside of $[x_L, x_R]$. $x_L$ is drawn uniformly from $[0.1, 0.45)$ and $x_R$ from $[0.55, 0.9)$. Lastly, the sign of $\mathbf{u}^0$ is flipped for all entries with a probability of 10%.

## B.2 KURAMOTO-SIVASHINSKY (KS)

The 1D KS equation reads as

$$\partial_t u + u\partial_x u + \partial_{xx} u + \nu\partial_{xxxx} u = 0 \quad x \in [0, L]. \tag{9}$$

The ICs are generated using the superposition of sinusoidal waves (Eq. (8)), but $k_i$ is sampled from $[1, 10)$, $A_i$ from $[-1, 1)$ and $\phi_i$ from $[0, 2\pi)$. No windowing or sign flips are applied. The total number of waves $N_w$ in this case is set to 10. Since we cannot omit the first part of the simulations as Lippe et al. (2023), we reduce the simulation time to 40s, but allow for more variance in the ICs to reach the chaotic behavior easier by increasing the number of wave functions of the IC. The trajectories are obtained using JAX-CFD (Dresdner et al., 2023). The PDE parameters are drawn uniformly from their range (no logarithmic scale).

## B.3 COMBINED EQUATION (CE)

We adopt the *combined equation* albeit without the *forcing* term and the corresponding numerical solver from Brandstetter et al. (2021).

$$\partial_t u + \partial_x \left( \alpha u^2 - \beta\partial_x u + \gamma\partial_{xx} u \right) = 0 \tag{10}$$

As for the IC, the domain of $k_i$ is set to $[1, 3)$ and for $A_i$ it is set as $[-0.4, 0.4)$. The number of waves $N_w$ is set to 5, and no windowing or sign flips are applied either. The PDE parameters are also drawn uniformly from their range. Depending on the choice of the PDE coefficients $(\alpha, \beta, \gamma)$, this equation recovers the Heat (0, 1, 0), Burgers (0.5, 1, 0), or the Korteweg-de-Vries (3, 0, 1) PDE. The spatial domain is set to $x \in [0, 16]$.

## B.4 COMPRESSIBLE NAVIER-STOKES (CNS)

The CNS equations from PDEBench (Takamoto et al., 2022) are written as

$$\partial_t \rho + \nabla \cdot (\rho\mathbf{v}) = 0, \tag{11a}$$

$$\rho(\partial_t \mathbf{v} + \mathbf{v} \cdot \nabla\mathbf{v}) = -\nabla p + \eta\triangle\mathbf{v} + (\zeta + \eta/3)\nabla(\nabla \cdot \mathbf{v}), \tag{11b}$$

$$\partial_t(\epsilon + \rho v^2/2) + \nabla \cdot [(p + \epsilon + \rho v^2/2)\mathbf{v} - \mathbf{v} \cdot \sigma'] = \mathbf{0}, \tag{11c}$$

where $\sigma'$ is the viscous tensor. For 2D, the equation has four channels (density $\rho$, pressure $p$, velocity x-component $v_x$, and y-component $v_y$, whereas, for 3D, we have an extra velocity-z component $v_z$ as the fifth channel in addition to the above. The spatial domain is set to $\mathbf{x} \in [0, 1] \times [0, 1]$ for 2D, and as the unit cube ($[0, 1] \times [0, 1] \times [0, 1]$) for 3D. We use the JAX simulator and IC generator from PDEBench (Takamoto et al., 2022) for CNS equations. The PDE parameters are drawn in logarithmic scale as in Eq. (7). The IC generator for the pressure, density, and velocity channels is also based on the superposition of sinusoidal functions. However, the velocity channels are renormalized so that the IC has a given input Mach number. Secondly, we constrain the density channel to be positive by

$$\mathbf{u}_\rho = \rho_0(1 + \Delta_\rho\,\mathbf{u}'_\rho\,/\max_x(|\,\mathbf{u}'_\rho(x)|) \tag{12}$$

where $\rho_0$ is sampled from $[0.1, 10)$ and $\Delta_\rho$ from $[0.013, 0.26)$. The pressure channel $p$ is similarly transformed using $\Delta_p \in [0.04, 0.8)$. The offset $p_0$ is defined relatively to $\rho_0$ as $p_0 = T_0\rho_0$ with

$T_0 \in [0.1, 10)$. The compressibility is reduced using a Helmholtz-decomposition (Takamoto et al., 2022). A windowing is applied with a probability of $50\%$ to a channel. For 3D CNS, the considered domain for the PDE coefficients is $\eta, \zeta \in [10^{-3}, 10^{-1})$. For 1D CNS, the PDE coefficients are set to be equal and not independently drawn.

# C    ADDITIONAL MODEL AND TRAINING DETAILS

This section describes the baseline surrogate models used in more detail, lists the hyperparameters, and explains various training methods. First, we provide a short description of the base models used. Then, we explain the training methods and list the hyperparameters.

## C.1    FOURIER NEURAL OPERATORS (FNOS)

We use the FNO (Li et al., 2021b) implementation provided by PDEBench (Takamoto et al., 2022). FNOs are based on spectral convolutions, where the layer input is transformed using a Fast Fourier Transformation (FFT), multiplied in the Fourier space with a weight matrix, and then transformed back using an inverse FFT. Following the recent observations made in Lanthaler et al. (2023; 2024) that only a small fixed number of modes are sufficient to achieve the needed expressivity of FNO, we retain only a limited number of low-frequency Fourier modes and discard the ones with higher frequencies. The raw PDE parameter values are appended as additional constant channels to the model input (Takamoto et al., 2023) as the conditioning factor.

## C.2    U-SHAPED NETWORKS (U-NETS)

U-Net (Ronneberger et al., 2015) is a common architecture in computer vision, particularly for perception and semantic segmentation tasks. The structure resembles an hourglass, where the inputs are first successively downsampled at multiple levels and then gradually, with the same number of levels, upsampled back to the original input resolution. This structure allows the model to capture and process spatial information at multiple scales and resolutions. The U-Net used in this paper is based on the modern U-Net version of Gupta & Brandstetter (2023), which differs from the original U-Net (Ronneberger et al., 2015) by including improvements such as group normalization (Wu & He, 2018). The model is conditioned on the input PDE parameter values, where they are transformed into vectors using a learnable Fourier embedding (Vaswani et al., 2017) and a projection layer and are then added to the convolutional layers' inputs in the *up* and *down* blocks.

## C.3    SINENET

U-Nets were originally designed for semantic segmentation problems in medical images (Ronneberger et al., 2015). Due to its intrinsic capabilities for multi-scale representation modeling, U-Nets have been widely adopted by the SciML community for PDE solving (Takamoto et al., 2022; Gupta & Brandstetter, 2023; Lippe et al., 2023; Rahman et al., 2022; Ovadia et al., 2023). One of the important components of U-Nets to recover high-resolution details in the upsampling path is by the fusion of feature maps using skip connections. This does not cause an issue for semantic segmentation tasks since the desired output for a given image is a segmentation mask. However, in the context of time-dependent PDE solving, specifically for advection-type PDEs modeling transport phenomena, this is not well-suited since there will be a "lag" in the feature maps of the downsampling path since the upsampling path is expected to predict the solution **u** for the next timestep. This detail was overlooked in U-Net adaptations for time-dependent PDE solving. SineNet is a recently introduced image-to-image model that aims to mitigate this problem by stacking several U-Nets, called *waves*, drastically reducing the feature misalignments. More formally, SineNet learns the mapping

$$\boldsymbol{x}_t = P(\{\boldsymbol{u}_{t-h+1}, \ldots, \boldsymbol{u}_t\})$$

$$\boldsymbol{u}_{t+1} = Q(\boldsymbol{x}_{t+1})$$

$$\boldsymbol{x}_{t+\Delta_k} = V_k(\boldsymbol{x}_{t-\Delta_{k-1}}), \qquad k = 1, \ldots, K$$

Unlike the original SineNet, our adaptation uses only one temporal step as a context to predict the solution for the subsequent timestep.

## C.4 HYPERPARAMETERS AND TRAINING PROTOCOLS

During AL, we use $m = 1$ for power sampling and a prediction batch size of 200 for the pool, except for 3D CNS for which we use 16 due to memory limitations. The features of all inputs are projected using the sketch operator to a dimension of 512. Table 2 lists the model hyperparameters.

| U-Net | |
|---|---|
| Activation | GELU (Hendrycks & Gimpel, 2016) |
| Conditioning | Fourier (Vaswani et al., 2017) |
| Channel multiplier | [1, 2, 2, 4] |
| Hidden Channels | 16 |
| # Params | 3,378,865 (1D) / 9,182,036 (2D) |
| FNO | |
| Activation | GELU (Hendrycks & Gimpel, 2016) |
| Conditioning | Additional input channel |
| Layers | 4 |
| Width | 64 (1D) / 32 (2D & 3D) |
| Modes | 20 |
| # Params | 680,834 (1D) / 6,563,110 (2D) / 262,153,959 (3D) |
| SineNet | |
| Activation | GELU (Hendrycks & Gimpel, 2016) |
| Conditioning | Fourier (Vaswani et al., 2017) |
| Hidden Channels | 32 |
| Waves | 4 |
| # Params | 5,020,840 (2D) |

Table 2: Model hyperparameters. For FNO 1D, the parameter counts are for Burger's PDE, whereas for FNO 2D and 3D, the CNS equations of respective spatial dimensions are considered.

The inputs are channel-wise normalized using the standard deviation of the different channels on the initial data set. The outputs are denormalized accordingly. The input only consists of the current state $\mathbf{u}_t$, not including data from prior timesteps. All models are used to predict the difference to the current timestep (for U-Net, the outputs are multiplied with a fixed factor of 0.3 following Lippe et al. (2023)).

We employ one- and two-step training strategies during the training phase and a complete rollout of the trajectories during validation. For the FNO model in the 2D and 3D experiments, we found it better to use the teacher-forcing schedule from Takamoto et al. (2023). We found it necessary to add gradient clipping to prevent a sudden divergence in the training curve. To account for the very different gradient norms among problems, we set the upper limit to 5 times the highest gradient found in the first five epochs. Afterward, the limit is adapted using a moving average.

We conduct experiments on the time-dependent 3D spatial Navier-Stokes equations and evaluate the efficacy of active learning in this challenging setup. For 3D, we set the pool size for the active learning methods on 3D CNS as 30,000 due to memory and compute time limitations. We train the ML models, 3D FNO, using a teacher forcing schedule (Takamoto et al., 2023) with a batch size of 10 for four active learning iterations. The initial training data consists of 64 trajectories, whereas the validation and test sets each have 512 trajectories.

## C.5 HARDWARE AND RUNTIME

The experiments were performed on NVIDIA GeForce RTX 4090 GPUs (one per experiment), except for the 3D CNS case, which was performed on a single 96 GB H100 GPU. Table 3 shows the runtime and GPU memory required for the PDEs during training.

|          | Burgers | KS   | CE   | 1D CNS | 2D CNS | 3D CNS |
|----------|---------|------|------|--------|--------|--------|
| Runtime in h | | | | | | |
| Random   | 15.6    | 13.4 | 16.2 | 19.2   | 37.6   | 9.5    |
| SBAL     | 21.6    | 20.9 | 25.4 | 26.8   | 55.8   | 14.4   |
| LCMD     | 15.1    | 13.7 | 17.0 | 20.3   | 38.2   | 12.4   |
| Core-Set | 14.7    | 13.7 | 16.7 | 20.3   | 39.9   | 12.6   |
| Top-K    | 21.8    | 20.4 | 26.1 | 26.9   | 56.5   | 14.4   |
| BAIT     | 15.6    | 13.8 | 17.3 | 20.5   | 40.0   | 12.9   |
| LHS      | 18.0    | 13.4 | 16.8 | 23.7   | 39.8   | 9.6    |
| Training Memory in GB | | | | | | |
| All      | 8.16    | 8.18 | 4.47 | 6.88   | 7.29   | 66.63  |

Table 3: Total runtime of the different AL methods and the memory during training (since all methods train the same model, the memory usage during training is identical).

## C.6 TIMING EXPERIMENT

A realistic time measurement for the simulator of Burgers' equation is challenging. Firstly, we observed that we can reach the shortest time per trajectory by setting the batch size to 4096 (0.52 seconds). Therefore, we use this as the fixed time per trajectory. The actual simulation times per AL iteration are higher since we start with batch sizes below this saturation point. Secondly, the simulation step size is adapted to the PDE parameter value due to the CFL condition (Lewy et al., 1928). Therefore, it would be beneficial to batch similar parameter values together and also to consider the parameter simulation costs in the acquisition function. Figure 9 shows training, selection, and simulation times.

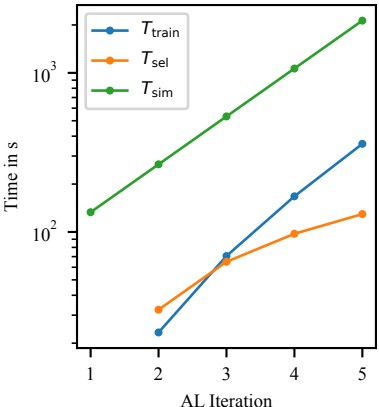

Figure 9: Cumulative training, selection, and simulation times necessary to reach the given active learning iteration (e.g., time to select data for iteration 2 counted in iteration 2) for 1D Burgers PDE.

The FNO surrogate used for selection is only trained for 20 epochs with a batch size of 1024. We use one-step training, and the learning rate of 0.001 is not annealed. The model itself has a width of 20 and uses 20 modes, resulting in 36,706 parameters. During selection, a batch size of 32,768 is used.

## D FRAMEWORK OVERVIEW

The framework has three major components: `Model`, `BatchSelection`, and `Task`. `Task` acts as a container of all the PDE-specific information and contains the `Simulator`, `PDEParamGenerator`, and `ICGenerator` classes. `PDEParamGenerator` and `ICGenerator` can draw samples from the test input distribution $p_T$. The inputs are first

drawn from a normalized range and then transformed into the actual inputs. Afterward, the inputs can be passed to the simulator to be evolved into a trajectory. Listing 1 shows the pseudocode of the (random) data generation pipeline. In order to implement a new PDE, a user has to implement a new subclass of `Simulator` overwrite the `__call__` function and, if desired, add a new `ICGenerator`.

```python
class PDEParamGenerator:

    def get_normed_pde_params(self, n):
        # Generates the random PDE parameters in a normed space
        # (e.g. between 0 and 1).

    def get_pde_params(self, pde_params_normed):
        # Transforms the normed parameters to their true value.

class ICGenerator:

    def initialize_ic_params(self, n):
        # Generates the random parameters of an IC (e.g. Mach number).

    def generate_initial_conditions(self, ic_params, pde_params)
        # Transforms the IC parameters and PDE parameters to the IC.

class Simulator:

    def __call__(self, ic, pde_params, grid):
        # Evolves the IC for a given PDE parameter.

# generate pde parameters
pde_params_normed = pde_gen.get_normed_pde_params(n)
pde_params = pde_gen.get_pde_params(pde_params_normed)

# generate ICs
ic_params = ic_gen.initialize_ic_params(n)
ic_gen.generate_initial_conditions(ic_params, pde_params)

trajectories = sim(ic, pde_param, grid)
```

Listing 1: Interface and example code for generating inputs and simulation.

Listing 2 shows the interface for the `Model` and `ProbModel` classes. `Model` provides functions to rollout a surrogate and deals with the training and evaluation. In order to add a new surrogate, a user has to overwrite the `forward` method. The `rollout` function also allows to get the internal model features for distance-based acquisition functions. `ProbModel` is an extension of the `Model` class, which adds the possibility of getting an uncertainty estimate. After training the model, the `BatchSelection` class is called in order to select a new set of inputs. The most important subclass is the `PoolBased` class, which deals with managing the pool and provides the `select_next` method, which a new pool-based method has to overwrite.

```python
class Model(nn.Module):

    def init_training(self, al_iter):
        # Reset model, optimizer, scheduler, ...

    def forward(self, xx, grid, param, return_features):
        # Predict the next state.

    def rollout(self, xx, grid, final_step, param,  return_features):
        # Autoregressive rollout of the model until timestep final_step.

    def evaluate(self, step, loader, prefix):
        # Evaluate the model on the given dataset (e.g. test).

    def train_single_epoch(self, current_epoch, total_epoch, num_epoch):
        # Train the model for one epoch.

    def train_n_epoch(self, al_iter, num_epoch):
        # Train the model.

class ProbModel(Model):

    def uncertainty(self, xx, grid, param):
        # Get uncertainty over the next state.

    def unc_roll_out(self, xx, grid, final_step, param, return_features):
        # Compute prediction and uncertainty of the rollout.

class BatchSelection:

    def generate(self, prob_model, al_iter, train_loader):
        # Select new inputs and pass them to the simulator.

class PoolBased(BatchSelection):

    def select_next(self, step, prob_model, ic_pool, pde_param_pool,
        ic_train, pde_param_train, grid, al_iter):
        # Select new input from (ic_pool, pde_param_pool).

for al_iter in range(num_al_iter):
    # retrain model
    prob_model.train_n_epoch(al_iter, num_epoch)

    # select next inputs
    batch_sel.generate(prob_model, al_iter, train_loader)
```

Listing 2: Interface and example code for the neural operator models and AL methods.

# E    ADDITIONAL EXPERIMENTS

Figure 10 shows an experiment with a smaller pool that is completely labeled after the last AL iteration. Sufficient pool size is important for the AL algorithm in order to be able to focus on the difficult dynamical regions.

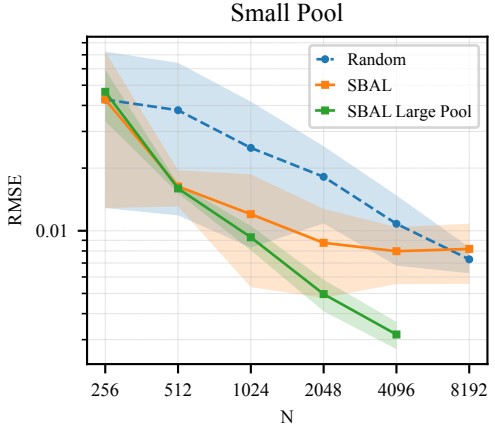

Figure 10: Error over the number of trajectories for an experiment with a pool size of only 8192 possible inputs (including initial data) on CE. Compared to the main SBAL results with the larger pool of 100,000 samples.

# F    DETAILED RESULTS

Tables 4-9 list the results from the main experiments. Table 10 shows the Pearson and Spearman coefficient of the average uncertainty per trajectory with the average error per trajectory. Among the PDEs, the Pearson correlation coefficient is the lowest on CE. The Spearman coefficient, which measures the correlation in terms of the ranking, is above 0.54 on average for all experiments.

| Iteration | 1 | 2 | 3 | 4 | 5 |
|---|---|---|---|---|---|
| RMSE $\times 10^{-2}$ | | | | | |
| Random | $3.684 \pm 1.203$ | $3.278 \pm 2.107$ | $1.607 \pm 0.485$ | $1.062 \pm 0.614$ | $0.552 \pm 0.133$ |
| SBAL | $3.684 \pm 1.203$ | $1.179 \pm 0.223$ | $0.586 \pm 0.106$ | $0.400 \pm 0.075$ | $\mathbf{0.259 \pm 0.028}$ |
| LCMD | $3.684 \pm 1.203$ | $\mathbf{0.808 \pm 0.053}$ | $\mathbf{0.521 \pm 0.052}$ | $0.394 \pm 0.043$ | $0.269 \pm 0.014$ |
| Core-Set | $3.684 \pm 1.203$ | $1.021 \pm 0.160$ | $0.659 \pm 0.100$ | $0.476 \pm 0.134$ | $0.292 \pm 0.015$ |
| Top-K | $3.684 \pm 1.203$ | $1.494 \pm 0.250$ | $0.964 \pm 0.258$ | $0.477 \pm 0.044$ | $0.360 \pm 0.096$ |
| BAIT | $3.684 \pm 1.203$ | $0.903 \pm 0.138$ | $0.537 \pm 0.030$ | $\mathbf{0.392 \pm 0.035}$ | $0.266 \pm 0.024$ |
| LHS | $3.441 \pm 1.708$ | $1.930 \pm 0.300$ | $1.354 \pm 0.529$ | $1.057 \pm 0.539$ | $0.521 \pm 0.117$ |
| 50% Quantile $\times 10^{-2}$ | | | | | |
| Random | $0.182 \pm 0.015$ | $0.122 \pm 0.015$ | $0.083 \pm 0.010$ | $\mathbf{0.058 \pm 0.005}$ | $\mathbf{0.044 \pm 0.007}$ |
| SBAL | $0.182 \pm 0.015$ | $0.178 \pm 0.032$ | $0.105 \pm 0.011$ | $0.078 \pm 0.011$ | $0.054 \pm 0.006$ |
| LCMD | $0.182 \pm 0.015$ | $0.129 \pm 0.014$ | $0.101 \pm 0.015$ | $0.068 \pm 0.008$ | $0.050 \pm 0.006$ |
| Core-Set | $0.182 \pm 0.015$ | $0.169 \pm 0.017$ | $0.133 \pm 0.013$ | $0.094 \pm 0.014$ | $0.063 \pm 0.008$ |
| Top-K | $0.182 \pm 0.015$ | $0.197 \pm 0.020$ | $0.176 \pm 0.024$ | $0.109 \pm 0.010$ | $0.078 \pm 0.012$ |
| BAIT | $0.182 \pm 0.015$ | $0.150 \pm 0.014$ | $0.115 \pm 0.011$ | $0.079 \pm 0.006$ | $0.058 \pm 0.008$ |
| LHS | $0.174 \pm 0.014$ | $\mathbf{0.116 \pm 0.014}$ | $\mathbf{0.081 \pm 0.009}$ | $0.062 \pm 0.007$ | $0.054 \pm 0.011$ |
| 95% Quantile $\times 10^{-2}$ | | | | | |
| Random | $1.468 \pm 0.136$ | $0.834 \pm 0.125$ | $0.502 \pm 0.037$ | $\mathbf{0.343 \pm 0.014}$ | $0.255 \pm 0.025$ |
| SBAL | $1.468 \pm 0.136$ | $1.054 \pm 0.248$ | $0.544 \pm 0.065$ | $0.409 \pm 0.064$ | $0.269 \pm 0.026$ |
| LCMD | $1.468 \pm 0.136$ | $\mathbf{0.669 \pm 0.069}$ | $0.503 \pm 0.091$ | $0.347 \pm 0.030$ | $0.259 \pm 0.020$ |
| Core-Set | $1.468 \pm 0.136$ | $0.865 \pm 0.123$ | $0.662 \pm 0.090$ | $0.503 \pm 0.113$ | $0.336 \pm 0.034$ |
| Top-K | $1.468 \pm 0.136$ | $1.273 \pm 0.177$ | $1.045 \pm 0.200$ | $0.575 \pm 0.064$ | $0.449 \pm 0.077$ |
| BAIT | $1.468 \pm 0.136$ | $0.800 \pm 0.160$ | $0.532 \pm 0.045$ | $0.378 \pm 0.021$ | $0.274 \pm 0.026$ |
| LHS | $1.390 \pm 0.142$ | $0.803 \pm 0.114$ | $\mathbf{0.474 \pm 0.038}$ | $0.344 \pm 0.027$ | $\mathbf{0.246 \pm 0.024}$ |
| 99% Quantile $\times 10^{-2}$ | | | | | |
| Random | $6.315 \pm 0.838$ | $3.327 \pm 0.724$ | $1.653 \pm 0.111$ | $0.968 \pm 0.046$ | $0.649 \pm 0.027$ |
| SBAL | $6.315 \pm 0.838$ | $3.169 \pm 0.945$ | $1.360 \pm 0.213$ | $0.987 \pm 0.239$ | $0.599 \pm 0.056$ |
| LCMD | $6.315 \pm 0.838$ | $\mathbf{1.802 \pm 0.157}$ | $\mathbf{1.223 \pm 0.237}$ | $\mathbf{0.819 \pm 0.108}$ | $\mathbf{0.573 \pm 0.041}$ |
| Core-Set | $6.315 \pm 0.838$ | $2.461 \pm 0.500$ | $1.756 \pm 0.360$ | $1.153 \pm 0.295$ | $0.703 \pm 0.056$ |
| Top-K | $6.315 \pm 0.838$ | $4.456 \pm 1.685$ | $3.251 \pm 1.039$ | $1.347 \pm 0.129$ | $1.048 \pm 0.326$ |
| BAIT | $6.315 \pm 0.838$ | $2.371 \pm 0.718$ | $1.255 \pm 0.108$ | $0.853 \pm 0.065$ | $0.612 \pm 0.055$ |
| LHS | $6.215 \pm 1.012$ | $3.017 \pm 0.476$ | $1.515 \pm 0.109$ | $0.963 \pm 0.046$ | $0.650 \pm 0.047$ |

Table 4: Error metrics on Burgers' equation.

| Iteration | 1 | 2 | 3 | 4 | 5 |
|---|---|---|---|---|---|
| | | | RMSE | | |
| Random | $0.452 \pm 0.026$ | $0.370 \pm 0.012$ | $0.312 \pm 0.013$ | $0.272 \pm 0.010$ | $0.229 \pm 0.010$ |
| SBAL | $0.452 \pm 0.026$ | $\mathbf{0.347 \pm 0.020}$ | $\mathbf{0.281 \pm 0.010}$ | $\mathbf{0.236 \pm 0.008}$ | $\mathbf{0.200 \pm 0.012}$ |
| LCMD | $0.452 \pm 0.026$ | $0.370 \pm 0.009$ | $0.315 \pm 0.013$ | $0.266 \pm 0.019$ | $0.219 \pm 0.018$ |
| Core-Set | $0.452 \pm 0.026$ | $0.389 \pm 0.011$ | $0.335 \pm 0.013$ | $0.278 \pm 0.006$ | $0.235 \pm 0.020$ |
| Top-K | $0.452 \pm 0.026$ | $0.378 \pm 0.018$ | $0.305 \pm 0.011$ | $0.264 \pm 0.014$ | $0.225 \pm 0.015$ |
| BAIT | $0.452 \pm 0.026$ | $\underline{0.368 \pm 0.017}$ | $\underline{0.294 \pm 0.016}$ | $\underline{0.240 \pm 0.009}$ | $\underline{0.205 \pm 0.011}$ |
| LHS | $0.439 \pm 0.008$ | $0.369 \pm 0.024$ | $0.316 \pm 0.011$ | $0.270 \pm 0.009$ | $0.222 \pm 0.012$ |
| | | | 50% Quantile | | |
| Random | $0.021 \pm 0.005$ | $\mathbf{0.011 \pm 0.002}$ | $\underline{0.008 \pm 0.001}$ | $\mathbf{0.005 \pm 0.001}$ | $0.003 \pm 0.001$ |
| SBAL | $0.021 \pm 0.005$ | $0.016 \pm 0.004$ | $0.013 \pm 0.003$ | $0.008 \pm 0.001$ | $0.006 \pm 0.001$ |
| LCMD | $0.021 \pm 0.005$ | $0.020 \pm 0.003$ | $0.016 \pm 0.003$ | $0.009 \pm 0.003$ | $0.006 \pm 0.001$ |
| Core-Set | $0.021 \pm 0.005$ | $0.022 \pm 0.003$ | $0.021 \pm 0.002$ | $0.014 \pm 0.002$ | $0.009 \pm 0.002$ |
| Top-K | $0.021 \pm 0.005$ | $0.020 \pm 0.003$ | $0.018 \pm 0.002$ | $0.012 \pm 0.003$ | $0.010 \pm 0.002$ |
| BAIT | $0.021 \pm 0.005$ | $0.020 \pm 0.003$ | $0.015 \pm 0.003$ | $0.008 \pm 0.001$ | $0.005 \pm 0.001$ |
| LHS | $0.019 \pm 0.001$ | $\underline{0.011 \pm 0.002}$ | $\mathbf{0.007 \pm 0.001}$ | $\underline{0.005 \pm 0.001}$ | $\mathbf{0.003 \pm 0.001}$ |
| | | | 95% Quantile | | |
| Random | $0.603 \pm 0.106$ | $\underline{0.363 \pm 0.020}$ | $\mathbf{0.231 \pm 0.024}$ | $\mathbf{0.143 \pm 0.011}$ | $0.094 \pm 0.006$ |
| SBAL | $0.603 \pm 0.106$ | $0.376 \pm 0.060$ | $0.255 \pm 0.031$ | $0.163 \pm 0.022$ | $0.119 \pm 0.018$ |
| LCMD | $0.603 \pm 0.106$ | $0.458 \pm 0.024$ | $0.344 \pm 0.024$ | $0.230 \pm 0.035$ | $0.140 \pm 0.023$ |
| Core-Set | $0.603 \pm 0.106$ | $0.501 \pm 0.025$ | $0.425 \pm 0.034$ | $0.295 \pm 0.021$ | $0.213 \pm 0.053$ |
| Top-K | $0.603 \pm 0.106$ | $0.458 \pm 0.017$ | $0.340 \pm 0.026$ | $0.257 \pm 0.039$ | $0.188 \pm 0.016$ |
| BAIT | $0.603 \pm 0.106$ | $0.450 \pm 0.051$ | $0.269 \pm 0.043$ | $0.163 \pm 0.020$ | $0.100 \pm 0.012$ |
| LHS | $0.572 \pm 0.020$ | $\mathbf{0.352 \pm 0.065}$ | $0.238 \pm 0.027$ | $\underline{0.148 \pm 0.016}$ | $\mathbf{0.091 \pm 0.006}$ |
| | | | 99% Quantile | | |
| Random | $2.368 \pm 0.153$ | $1.844 \pm 0.105$ | $1.382 \pm 0.117$ | $1.040 \pm 0.092$ | $0.708 \pm 0.048$ |
| SBAL | $2.368 \pm 0.153$ | $\mathbf{1.655 \pm 0.137}$ | $\mathbf{1.177 \pm 0.100}$ | $\mathbf{0.844 \pm 0.103}$ | $0.619 \pm 0.093$ |
| LCMD | $2.368 \pm 0.153$ | $1.811 \pm 0.056$ | $1.440 \pm 0.097$ | $1.151 \pm 0.123$ | $0.802 \pm 0.149$ |
| Core-Set | $2.368 \pm 0.153$ | $1.920 \pm 0.077$ | $1.571 \pm 0.090$ | $1.230 \pm 0.046$ | $0.982 \pm 0.202$ |
| Top-K | $2.368 \pm 0.153$ | $1.860 \pm 0.126$ | $1.356 \pm 0.092$ | $1.138 \pm 0.086$ | $0.873 \pm 0.119$ |
| BAIT | $2.368 \pm 0.153$ | $\underline{1.782 \pm 0.112}$ | $\underline{1.265 \pm 0.131}$ | $\underline{0.863 \pm 0.051}$ | $\mathbf{0.607 \pm 0.055}$ |
| LHS | $2.296 \pm 0.053$ | $1.844 \pm 0.160$ | $1.426 \pm 0.089$ | $1.036 \pm 0.082$ | $0.667 \pm 0.058$ |

Table 5: Error metrics on KS.

| Iteration | 1 | 2 | 3 | 4 | 5 |
|---|---|---|---|---|---|
| | | | RMSE $\times 10^{-2}$ | | |
| Random | $4.651 \pm 1.293$ | $3.814 \pm 1.121$ | $2.609 \pm 0.466$ | $1.630 \pm 0.257$ | $1.108 \pm 0.117$ |
| SBAL | $4.651 \pm 1.293$ | $1.597 \pm 0.083$ | $0.931 \pm 0.125$ | $\mathbf{0.496 \pm 0.087}$ | $\mathbf{0.318 \pm 0.048}$ |
| LCMD | $4.651 \pm 1.293$ | $\underline{1.528 \pm 0.121}$ | $0.957 \pm 0.114$ | $0.609 \pm 0.107$ | $\underline{0.338 \pm 0.041}$ |
| Core-Set | $4.651 \pm 1.293$ | $1.596 \pm 0.235$ | $1.033 \pm 0.076$ | $0.761 \pm 0.230$ | $0.424 \pm 0.053$ |
| Top-K | $4.651 \pm 1.293$ | $1.678 \pm 0.099$ | $\underline{0.904 \pm 0.101}$ | $\underline{0.529 \pm 0.103}$ | $0.373 \pm 0.077$ |
| BAIT | $4.651 \pm 1.293$ | $\mathbf{1.415 \pm 0.187}$ | $\mathbf{0.900 \pm 0.102}$ | $0.660 \pm 0.159$ | $0.424 \pm 0.124$ |
| LHS | $5.130 \pm 0.808$ | $3.626 \pm 1.011$ | $2.668 \pm 0.383$ | $1.852 \pm 0.301$ | $1.312 \pm 0.144$ |
| | | | 50% Quantile $\times 10^{-2}$ | | |
| Random | $0.238 \pm 0.025$ | $\underline{0.166 \pm 0.036}$ | $\underline{0.125 \pm 0.021}$ | $0.083 \pm 0.005$ | $0.065 \pm 0.004$ |
| SBAL | $0.238 \pm 0.025$ | $0.200 \pm 0.024$ | $0.125 \pm 0.009$ | $\mathbf{0.076 \pm 0.008}$ | $\mathbf{0.052 \pm 0.004}$ |
| LCMD | $0.238 \pm 0.025$ | $0.171 \pm 0.007$ | $0.128 \pm 0.015$ | $\underline{0.083 \pm 0.008}$ | $\underline{0.054 \pm 0.004}$ |
| Core-Set | $0.238 \pm 0.025$ | $0.224 \pm 0.070$ | $0.168 \pm 0.020$ | $0.143 \pm 0.059$ | $0.083 \pm 0.009$ |
| Top-K | $0.238 \pm 0.025$ | $0.211 \pm 0.019$ | $0.155 \pm 0.016$ | $0.111 \pm 0.015$ | $0.073 \pm 0.008$ |
| BAIT | $0.238 \pm 0.025$ | $0.186 \pm 0.018$ | $0.146 \pm 0.011$ | $0.108 \pm 0.011$ | $0.080 \pm 0.006$ |
| LHS | $0.249 \pm 0.030$ | $\mathbf{0.145 \pm 0.022}$ | $\mathbf{0.117 \pm 0.019}$ | $0.085 \pm 0.011$ | $0.066 \pm 0.003$ |
| | | | 95% Quantile $\times 10^{-2}$ | | |
| Random | $2.373 \pm 0.220$ | $1.619 \pm 0.222$ | $1.090 \pm 0.050$ | $0.695 \pm 0.039$ | $0.516 \pm 0.019$ |
| SBAL | $2.373 \pm 0.220$ | $1.723 \pm 0.126$ | $\mathbf{0.980 \pm 0.070}$ | $\mathbf{0.510 \pm 0.036}$ | $\mathbf{0.313 \pm 0.014}$ |
| LCMD | $2.373 \pm 0.220$ | $\mathbf{1.485 \pm 0.121}$ | $\underline{1.038 \pm 0.087}$ | $\underline{0.609 \pm 0.061}$ | $0.361 \pm 0.020$ |
| Core-Set | $2.373 \pm 0.220$ | $1.902 \pm 0.379$ | $1.389 \pm 0.126$ | $1.102 \pm 0.469$ | $0.598 \pm 0.095$ |
| Top-K | $2.373 \pm 0.220$ | $1.901 \pm 0.100$ | $1.236 \pm 0.099$ | $0.739 \pm 0.151$ | $0.416 \pm 0.039$ |
| BAIT | $2.373 \pm 0.220$ | $1.567 \pm 0.152$ | $1.121 \pm 0.085$ | $0.753 \pm 0.075$ | $0.515 \pm 0.047$ |
| LHS | $2.537 \pm 0.213$ | $\underline{1.516 \pm 0.098}$ | $1.080 \pm 0.098$ | $0.709 \pm 0.057$ | $0.530 \pm 0.013$ |
| | | | 99% Quantile $\times 10^{-2}$ | | |
| Random | $10.192 \pm 1.523$ | $7.260 \pm 1.226$ | $4.741 \pm 0.281$ | $2.893 \pm 0.227$ | $1.870 \pm 0.099$ |
| SBAL | $10.192 \pm 1.523$ | $4.756 \pm 0.215$ | $\mathbf{2.701 \pm 0.251}$ | $\mathbf{1.433 \pm 0.070}$ | $\mathbf{0.896 \pm 0.053}$ |
| LCMD | $10.192 \pm 1.523$ | $\mathbf{4.198 \pm 0.103}$ | $2.787 \pm 0.210$ | $\underline{1.631 \pm 0.178}$ | $0.991 \pm 0.038$ |
| Core-Set | $10.192 \pm 1.523$ | $5.056 \pm 0.827$ | $3.526 \pm 0.212$ | $2.638 \pm 1.069$ | $1.446 \pm 0.290$ |
| Top-K | $10.192 \pm 1.523$ | $5.382 \pm 0.373$ | $3.174 \pm 0.181$ | $1.756 \pm 0.448$ | $\underline{0.972 \pm 0.092}$ |
| BAIT | $10.192 \pm 1.523$ | $\underline{4.290 \pm 0.307}$ | $2.896 \pm 0.141$ | $1.939 \pm 0.172$ | $1.301 \pm 0.104$ |
| LHS | $10.785 \pm 1.740$ | $6.863 \pm 0.578$ | $4.778 \pm 0.272$ | $3.090 \pm 0.546$ | $1.874 \pm 0.056$ |

Table 6: Error metrics on CE.

| Iteration | 1 | 2 | 3 | 4 | 5 |
|---|---|---|---|---|---|
| | | | RMSE | | |
| Random | $3.054 \pm 1.276$ | $\mathbf{1.966 \pm 0.248}$ | $1.293 \pm 0.092$ | $0.997 \pm 0.079$ | $0.695 \pm 0.104$ |
| SBAL | $3.054 \pm 1.276$ | $2.093 \pm 0.380$ | $1.347 \pm 0.180$ | $0.867 \pm 0.097$ | $0.581 \pm 0.077$ |
| LCMD | $3.054 \pm 1.276$ | $2.291 \pm 1.030$ | $1.354 \pm 0.131$ | $\mathbf{0.856 \pm 0.092}$ | $\mathbf{0.555 \pm 0.049}$ |
| Core-Set | $3.054 \pm 1.276$ | $2.486 \pm 0.831$ | $1.922 \pm 0.583$ | $1.232 \pm 0.160$ | $0.753 \pm 0.227$ |
| Top-K | $3.054 \pm 1.276$ | $2.586 \pm 1.117$ | $1.467 \pm 0.158$ | $0.986 \pm 0.223$ | $0.618 \pm 0.087$ |
| BAIT | $3.054 \pm 1.276$ | $3.728 \pm 1.544$ | $1.649 \pm 0.262$ | $1.181 \pm 0.262$ | $0.599 \pm 0.072$ |
| LHS | $2.635 \pm 0.326$ | $2.159 \pm 0.480$ | $\mathbf{1.269 \pm 0.099}$ | $0.987 \pm 0.130$ | $0.669 \pm 0.055$ |
| | | | 50% Quantile | | |
| Random | $1.158 \pm 0.387$ | $\mathbf{0.758 \pm 0.122}$ | $\mathbf{0.479 \pm 0.044}$ | $\mathbf{0.347 \pm 0.043}$ | $0.206 \pm 0.019$ |
| SBAL | $1.158 \pm 0.387$ | $0.893 \pm 0.150$ | $0.589 \pm 0.084$ | $0.371 \pm 0.055$ | $0.231 \pm 0.046$ |
| LCMD | $1.158 \pm 0.387$ | $1.028 \pm 0.491$ | $0.604 \pm 0.065$ | $0.351 \pm 0.040$ | $\mathbf{0.196 \pm 0.017}$ |
| Core-Set | $1.158 \pm 0.387$ | $1.108 \pm 0.282$ | $0.963 \pm 0.301$ | $0.574 \pm 0.102$ | $0.313 \pm 0.093$ |
| Top-K | $1.158 \pm 0.387$ | $1.186 \pm 0.391$ | $0.702 \pm 0.069$ | $0.462 \pm 0.101$ | $0.272 \pm 0.047$ |
| BAIT | $1.158 \pm 0.387$ | $1.216 \pm 0.293$ | $0.758 \pm 0.105$ | $0.496 \pm 0.060$ | $0.256 \pm 0.047$ |
| LHS | $1.078 \pm 0.086$ | $0.832 \pm 0.207$ | $0.521 \pm 0.068$ | $0.348 \pm 0.080$ | $0.209 \pm 0.026$ |
| | | | 95% Quantile | | |
| Random | $6.356 \pm 3.150$ | $\mathbf{3.944 \pm 0.632}$ | $2.580 \pm 0.256$ | $1.942 \pm 0.145$ | $1.247 \pm 0.151$ |
| SBAL | $6.356 \pm 3.150$ | $4.236 \pm 0.939$ | $2.709 \pm 0.405$ | $1.696 \pm 0.250$ | $1.149 \pm 0.181$ |
| LCMD | $6.356 \pm 3.150$ | $4.774 \pm 2.267$ | $2.719 \pm 0.314$ | $\mathbf{1.688 \pm 0.209}$ | $\mathbf{1.055 \pm 0.102}$ |
| Core-Set | $6.356 \pm 3.150$ | $5.029 \pm 2.019$ | $3.998 \pm 1.348$ | $2.540 \pm 0.326$ | $1.519 \pm 0.523$ |
| Top-K | $6.356 \pm 3.150$ | $5.444 \pm 2.520$ | $3.022 \pm 0.373$ | $1.988 \pm 0.530$ | $1.225 \pm 0.189$ |
| BAIT | $6.356 \pm 3.150$ | $7.403 \pm 3.150$ | $3.439 \pm 0.543$ | $2.428 \pm 0.581$ | $1.170 \pm 0.156$ |
| LHS | $5.313 \pm 0.491$ | $4.399 \pm 1.146$ | $\mathbf{2.477 \pm 0.291}$ | $1.926 \pm 0.301$ | $1.267 \pm 0.151$ |
| | | | 99% Quantile | | |
| Random | $11.293 \pm 4.832$ | $7.296 \pm 0.897$ | $4.860 \pm 0.482$ | $3.687 \pm 0.291$ | $2.537 \pm 0.215$ |
| SBAL | $11.293 \pm 4.832$ | $\mathbf{7.290 \pm 1.582}$ | $4.720 \pm 0.761$ | $\mathbf{2.969 \pm 0.315}$ | $\mathbf{2.035 \pm 0.267}$ |
| LCMD | $11.293 \pm 4.832$ | $8.009 \pm 3.637$ | $4.651 \pm 0.522$ | $3.090 \pm 0.388$ | $2.095 \pm 0.253$ |
| Core-Set | $11.293 \pm 4.832$ | $8.403 \pm 2.948$ | $6.103 \pm 1.648$ | $4.191 \pm 0.546$ | $2.649 \pm 0.796$ |
| Top-K | $11.293 \pm 4.832$ | $8.677 \pm 4.075$ | $4.835 \pm 0.516$ | $3.305 \pm 0.719$ | $2.098 \pm 0.334$ |
| BAIT | $11.293 \pm 4.832$ | $14.736 \pm 6.750$ | $5.570 \pm 1.124$ | $4.182 \pm 1.347$ | $2.126 \pm 0.194$ |
| LHS | $9.637 \pm 1.813$ | $8.103 \pm 1.884$ | $\mathbf{4.582 \pm 0.325}$ | $3.596 \pm 0.326$ | $2.500 \pm 0.223$ |

Table 7: Error metrics on 1D CNS.

| Iteration | 1 | 2 | 3 | 4 | 5 |
|---|---|---|---|---|---|
| | | | RMSE | | |
| Random | $2.662 \pm 0.339$ | $2.162 \pm 0.029$ | $1.856 \pm 0.106$ | $1.572 \pm 0.072$ | $1.362 \pm 0.065$ |
| SBAL | $2.662 \pm 0.339$ | $\mathbf{1.979 \pm 0.226}$ | $1.790 \pm 0.203$ | $1.458 \pm 0.140$ | $\mathbf{1.205 \pm 0.027}$ |
| LCMD | $2.662 \pm 0.339$ | $\underline{1.991 \pm 0.293}$ | $1.734 \pm 0.189$ | $\mathbf{1.356 \pm 0.081}$ | $1.277 \pm 0.083$ |
| Core-Set | $2.662 \pm 0.339$ | $2.322 \pm 0.350$ | $\underline{1.731 \pm 0.168}$ | $1.613 \pm 0.202$ | $1.343 \pm 0.186$ |
| Top-K | $2.662 \pm 0.339$ | $2.684 \pm 1.129$ | $2.070 \pm 0.368$ | $1.623 \pm 0.524$ | $1.313 \pm 0.106$ |
| BAIT | $2.662 \pm 0.339$ | $2.167 \pm 0.164$ | $\mathbf{1.715 \pm 0.269}$ | $\underline{1.426 \pm 0.209}$ | $\underline{1.234 \pm 0.126}$ |
| LHS | $2.459 \pm 0.081$ | $2.134 \pm 0.148$ | $1.829 \pm 0.098$ | $1.514 \pm 0.059$ | $1.344 \pm 0.038$ |
| | | | 50% Quantile | | |
| Random | $0.506 \pm 0.119$ | $\mathbf{0.447 \pm 0.156}$ | $0.356 \pm 0.111$ | $0.266 \pm 0.087$ | $\mathbf{0.209 \pm 0.034}$ |
| SBAL | $0.506 \pm 0.119$ | $\underline{0.480 \pm 0.116}$ | $0.543 \pm 0.344$ | $0.336 \pm 0.063$ | $0.295 \pm 0.053$ |
| LCMD | $0.506 \pm 0.119$ | $0.574 \pm 0.361$ | $0.412 \pm 0.234$ | $0.317 \pm 0.065$ | $0.312 \pm 0.085$ |
| Core-Set | $0.506 \pm 0.119$ | $0.562 \pm 0.154$ | $0.411 \pm 0.085$ | $0.433 \pm 0.191$ | $0.408 \pm 0.120$ |
| Top-K | $0.506 \pm 0.119$ | $0.653 \pm 0.165$ | $0.521 \pm 0.133$ | $0.483 \pm 0.174$ | $0.400 \pm 0.065$ |
| BAIT | $0.506 \pm 0.119$ | $0.637 \pm 0.336$ | $0.392 \pm 0.076$ | $0.335 \pm 0.069$ | $0.311 \pm 0.093$ |
| LHS | $0.553 \pm 0.132$ | $0.503 \pm 0.068$ | $\mathbf{0.304 \pm 0.035}$ | $\mathbf{0.264 \pm 0.066}$ | $\underline{0.233 \pm 0.041}$ |
| | | | 95% Quantile | | |
| Random | $4.421 \pm 0.630$ | $3.491 \pm 0.154$ | $2.828 \pm 0.314$ | $2.317 \pm 0.207$ | $1.927 \pm 0.170$ |
| SBAL | $4.421 \pm 0.630$ | $3.308 \pm 0.550$ | $2.936 \pm 0.370$ | $2.310 \pm 0.349$ | $\mathbf{1.821 \pm 0.128}$ |
| LCMD | $4.421 \pm 0.630$ | $\mathbf{3.263 \pm 0.561}$ | $\mathbf{2.758 \pm 0.351}$ | $\mathbf{2.025 \pm 0.177}$ | $2.003 \pm 0.326$ |
| Core-Set | $4.421 \pm 0.630$ | $4.235 \pm 0.899$ | $2.952 \pm 0.375$ | $2.690 \pm 0.396$ | $2.189 \pm 0.437$ |
| Top-K | $4.421 \pm 0.630$ | $5.009 \pm 2.402$ | $3.891 \pm 0.921$ | $2.911 \pm 1.392$ | $2.238 \pm 0.289$ |
| BAIT | $4.421 \pm 0.630$ | $3.700 \pm 0.263$ | $\underline{2.783 \pm 0.547}$ | $\underline{2.238 \pm 0.404}$ | $\underline{1.900 \pm 0.273}$ |
| LHS | $4.173 \pm 0.299$ | $\underline{3.283 \pm 0.240}$ | $2.840 \pm 0.230$ | $2.250 \pm 0.102$ | $1.926 \pm 0.087$ |
| | | | 99% Quantile | | |
| Random | $11.378 \pm 1.863$ | $9.135 \pm 0.253$ | $7.754 \pm 0.507$ | $6.620 \pm 0.340$ | $5.735 \pm 0.320$ |
| SBAL | $11.378 \pm 1.863$ | $8.295 \pm 1.062$ | $\underline{7.195 \pm 0.786}$ | $6.058 \pm 0.573$ | $\mathbf{4.933 \pm 0.112}$ |
| LCMD | $11.378 \pm 1.863$ | $\mathbf{8.196 \pm 0.926}$ | $7.229 \pm 0.609$ | $\mathbf{5.569 \pm 0.362}$ | $5.265 \pm 0.399$ |
| Core-Set | $11.378 \pm 1.863$ | $9.739 \pm 1.416$ | $7.263 \pm 0.707$ | $6.646 \pm 0.794$ | $5.404 \pm 0.722$ |
| Top-K | $11.378 \pm 1.863$ | $11.424 \pm 5.585$ | $8.531 \pm 1.478$ | $6.466 \pm 2.101$ | $5.237 \pm 0.417$ |
| BAIT | $11.378 \pm 1.863$ | $8.948 \pm 0.487$ | $\mathbf{7.140 \pm 1.168}$ | $\underline{5.923 \pm 0.922}$ | $\underline{5.059 \pm 0.598}$ |
| LHS | $10.422 \pm 0.367$ | $8.800 \pm 0.769$ | $7.727 \pm 0.531$ | $6.374 \pm 0.198$ | $5.611 \pm 0.132$ |

Table 8: Error metrics on 2D CNS.

| Iteration | 1 | 2 | 3 | 4 |
|---|---|---|---|---|
| | | RMSE | | |
| Random | $3.159 \pm 0.915$ | $2.550 \pm 0.338$ | $2.364 \pm 0.429$ | $2.022 \pm 0.201$ |
| SBAL | $3.159 \pm 0.915$ | $2.265 \pm 0.402$ | $\underline{1.901 \pm 0.079}$ | $\underline{1.680 \pm 0.098}$ |
| LCMD | $3.159 \pm 0.915$ | $2.382 \pm 0.600$ | $2.011 \pm 0.222$ | $1.795 \pm 0.032$ |
| Core-Set | $3.159 \pm 0.915$ | $2.383 \pm 0.648$ | $\mathbf{1.890 \pm 0.199}$ | $\mathbf{1.632 \pm 0.197}$ |
| Top-K | $3.159 \pm 0.915$ | $2.635 \pm 0.216$ | $2.056 \pm 0.194$ | $1.823 \pm 0.231$ |
| BAIT | $3.159 \pm 0.915$ | $\mathbf{2.205 \pm 0.125}$ | $1.959 \pm 0.098$ | $1.697 \pm 0.133$ |
| LHS | $3.180 \pm 1.095$ | $2.567 \pm 0.648$ | $2.297 \pm 0.124$ | $2.017 \pm 0.226$ |
| | | 50% Quantile | | |
| Random | $1.092 \pm 0.187$ | $\underline{0.599 \pm 0.070}$ | $\mathbf{0.381 \pm 0.083}$ | $\mathbf{0.268 \pm 0.046}$ |
| SBAL | $1.092 \pm 0.187$ | $0.639 \pm 0.072$ | $0.420 \pm 0.048$ | $0.332 \pm 0.054$ |
| LCMD | $1.092 \pm 0.187$ | $0.862 \pm 0.435$ | $\underline{0.393 \pm 0.074}$ | $0.288 \pm 0.064$ |
| Core-Set | $1.092 \pm 0.187$ | $0.852 \pm 0.180$ | $0.455 \pm 0.114$ | $0.314 \pm 0.043$ |
| Top-K | $1.092 \pm 0.187$ | $0.952 \pm 0.404$ | $0.615 \pm 0.138$ | $0.598 \pm 0.285$ |
| BAIT | $1.092 \pm 0.187$ | $0.731 \pm 0.424$ | $0.510 \pm 0.408$ | $0.303 \pm 0.033$ |
| LHS | $1.090 \pm 0.215$ | $\mathbf{0.585 \pm 0.159}$ | $0.448 \pm 0.163$ | $\underline{0.272 \pm 0.050}$ |
| | | 95% Quantile | | |
| Random | $5.519 \pm 0.857$ | $4.255 \pm 0.480$ | $3.885 \pm 0.682$ | $3.162 \pm 0.150$ |
| SBAL | $5.519 \pm 0.857$ | $\underline{4.058 \pm 0.873}$ | $\mathbf{3.215 \pm 0.143}$ | $\mathbf{2.725 \pm 0.442}$ |
| LCMD | $5.519 \pm 0.857$ | $4.289 \pm 1.636$ | $\underline{3.433 \pm 0.375}$ | $3.020 \pm 0.262$ |
| Core-Set | $5.519 \pm 0.857$ | $4.520 \pm 1.904$ | $3.549 \pm 0.833$ | $\underline{2.806 \pm 0.841}$ |
| Top-K | $5.519 \pm 0.857$ | $4.935 \pm 1.120$ | $3.878 \pm 0.720$ | $3.275 \pm 0.424$ |
| BAIT | $5.519 \pm 0.857$ | $\mathbf{3.945 \pm 0.257}$ | $3.457 \pm 0.425$ | $2.831 \pm 0.416$ |
| LHS | $5.626 \pm 1.988$ | $4.108 \pm 0.743$ | $3.872 \pm 0.178$ | $3.157 \pm 0.449$ |
| | | 99% Quantile | | |
| Random | $12.929 \pm 6.254$ | $11.013 \pm 1.485$ | $10.187 \pm 2.298$ | $8.527 \pm 0.916$ |
| SBAL | $12.929 \pm 6.254$ | $9.128 \pm 2.003$ | $\underline{7.844 \pm 0.410}$ | $\underline{6.787 \pm 0.209}$ |
| LCMD | $12.929 \pm 6.254$ | $9.245 \pm 2.225$ | $8.367 \pm 0.805$ | $7.491 \pm 0.183$ |
| Core-Set | $12.929 \pm 6.254$ | $9.062 \pm 2.095$ | $\mathbf{7.649 \pm 0.618}$ | $\mathbf{6.713 \pm 0.686}$ |
| Top-K | $12.929 \pm 6.254$ | $9.744 \pm 2.160$ | $8.253 \pm 0.703$ | $7.072 \pm 0.737$ |
| BAIT | $12.929 \pm 6.254$ | $\mathbf{8.710 \pm 0.265}$ | $8.022 \pm 0.419$ | $7.073 \pm 0.532$ |
| LHS | $13.314 \pm 6.107$ | $11.343 \pm 4.375$ | $9.730 \pm 0.983$ | $8.411 \pm 1.143$ |

Table 9: Error metrics on 3D CNS.

| Iteration | 1 | 2 | 3 | 4 |
|---|---|---|---|---|
| | | Pearson | | |
| KS | $87.1 \pm 3.8$ | $84.9 \pm 2.3$ | $78.0 \pm 5.4$ | $80.5 \pm 3.7$ |
| CE | $49.2 \pm 16.2$ | $62.0 \pm 14.6$ | $41.3 \pm 22.1$ | $73.8 \pm 20.9$ |
| 1D CNS | $49.7 \pm 14.8$ | $67.0 \pm 22.3$ | $59.2 \pm 6.1$ | $55.6 \pm 5.1$ |
| 2D CNS | $78.2 \pm 6.4$ | $78.9 \pm 18.0$ | $90.8 \pm 2.7$ | $94.3 \pm 2.0$ |
| 3D CNS | $41.4 \pm 3.7$ | $65.5 \pm 11.9$ | $74.9 \pm 2.6$ | |
| Burgers $M = 2$ | $92.0 \pm 6.3$ | $71.3 \pm 27.1$ | $71.4 \pm 11.5$ | $67.9 \pm 18.4$ |
| Burgers $M = 6$ | $89.5 \pm 8.2$ | $60.9 \pm 26.7$ | $67.9 \pm 19.6$ | |
| | | Spearman | | |
| KS | $86.4 \pm 2.8$ | $83.0 \pm 2.8$ | $83.9 \pm 4.2$ | $82.7 \pm 0.4$ |
| CE | $87.4 \pm 1.7$ | $83.9 \pm 2.1$ | $81.2 \pm 1.0$ | $80.5 \pm 1.5$ |
| 1D CNS | $71.5 \pm 9.2$ | $54.4 \pm 14.3$ | $66.4 \pm 5.1$ | $68.1 \pm 3.2$ |
| 2D CNS | $94.6 \pm 2.4$ | $93.4 \pm 2.3$ | $91.1 \pm 3.9$ | $93.4 \pm 1.6$ |
| 3D CNS | $72.7 \pm 3.1$ | $86.0 \pm 0.6$ | $91.6 \pm 0.8$ | |
| Burgers $M = 2$ | $87.5 \pm 2.7$ | $83.2 \pm 11.0$ | $75.2 \pm 5.0$ | $73.7 \pm 5.3$ |
| Burgers $M = 6$ | $90.3 \pm 0.9$ | $84.5 \pm 2.3$ | $80.8 \pm 2.2$ | |

Table 10: Correlation coefficients in percent between the error and the uncertainty averages per trajectory, including the standard deviation. Computed for SBAL on the main experiments as well as the ensemble size ablation experiment.

## G OVERVIEW OF THE GENERATED DATASETS

In the following sections, we show visual examples of the data selected by random sampling and SBAL, and the marginal distributions of all PDE and IC parameters afterwards.

### G.1 EXAMPLE TRAJECTORIES

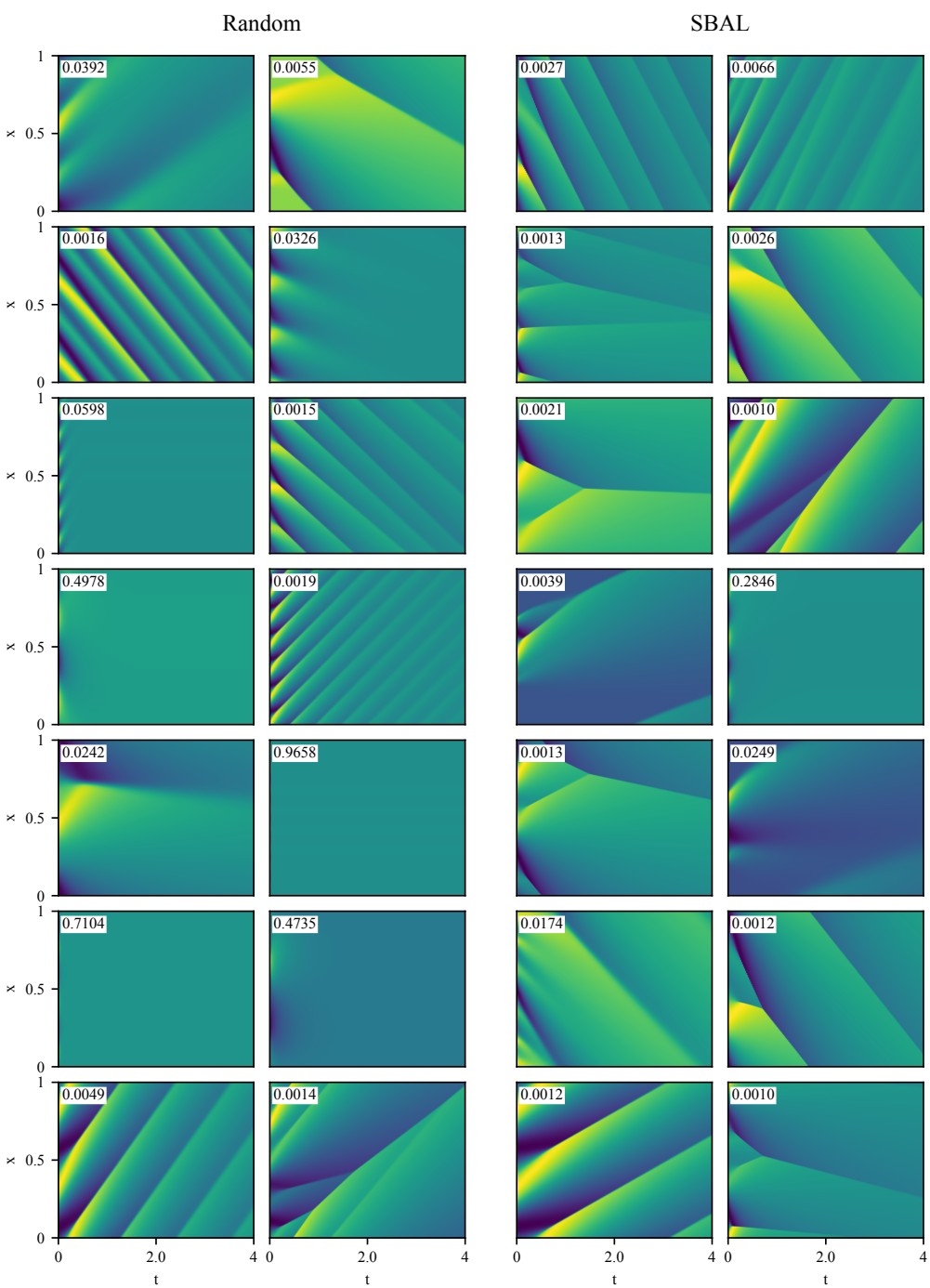

Figure 11: Example ground truth trajectories of random and SBAL on Burgers. The number on the top left of the trajectories shows the PDE parameter $\nu$.

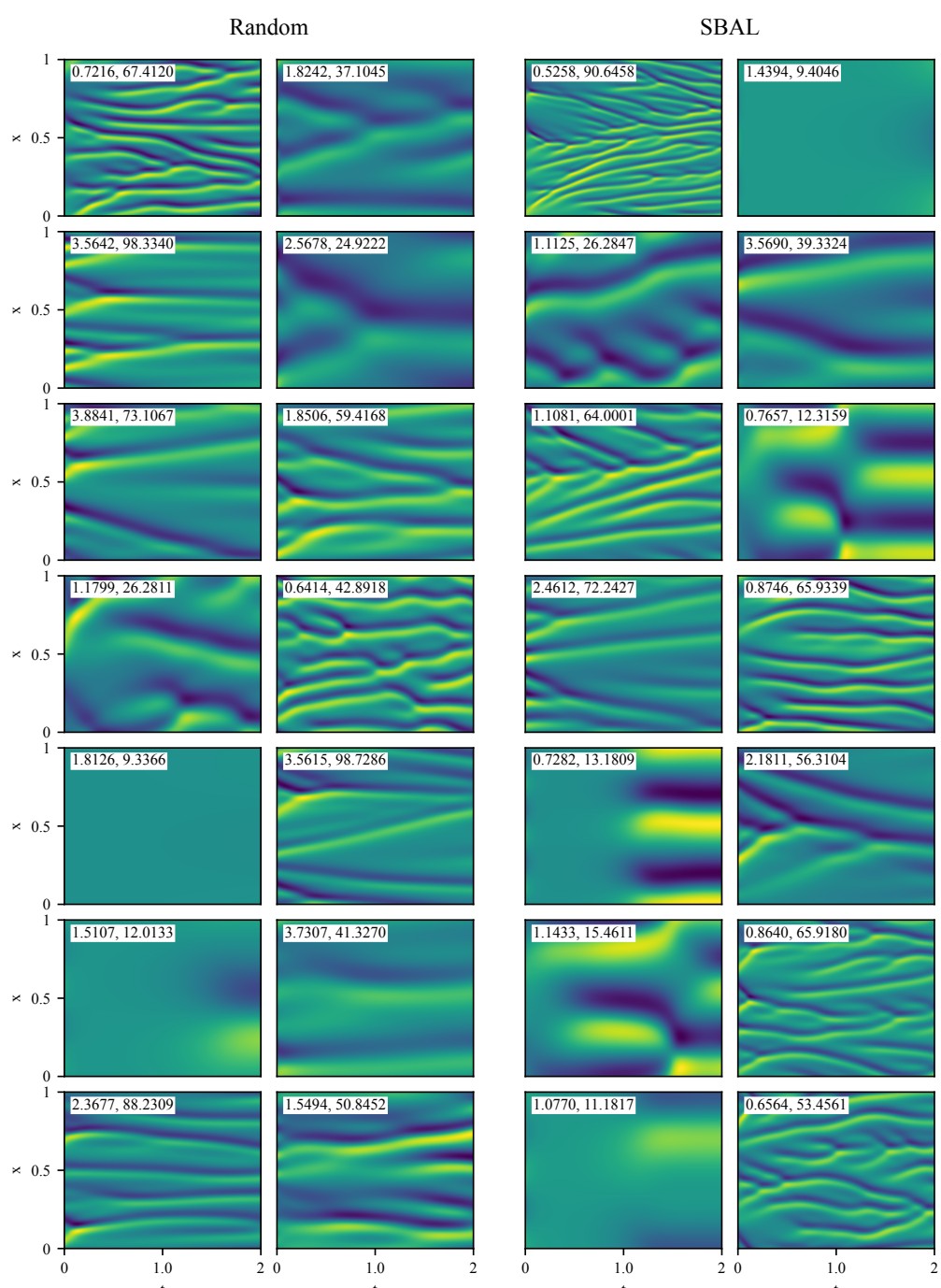

Figure 12: Example ground truth trajectories of random and SBAL on **KS**. The number on the top left of the trajectories shows the parameters $(\nu, L)$. The x-axis is shown in normalized values between 0 and 1 independent of the variable domain length $L$.

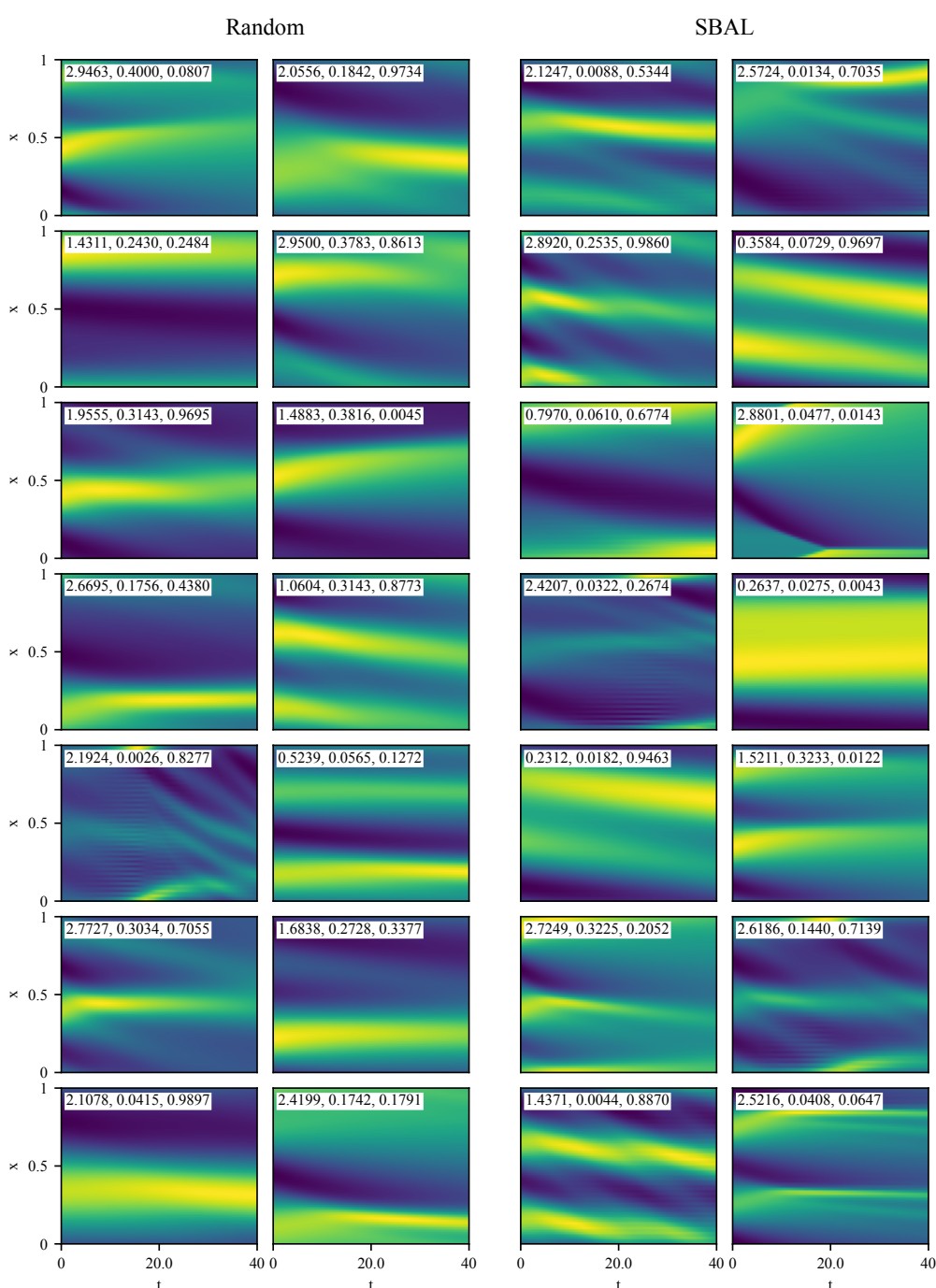

Figure 13: Example ground truth trajectories of random and SBAL on CE. The numbers on the top left of the trajectories show the PDE parameters ($\alpha$, $\beta$, $\gamma$).

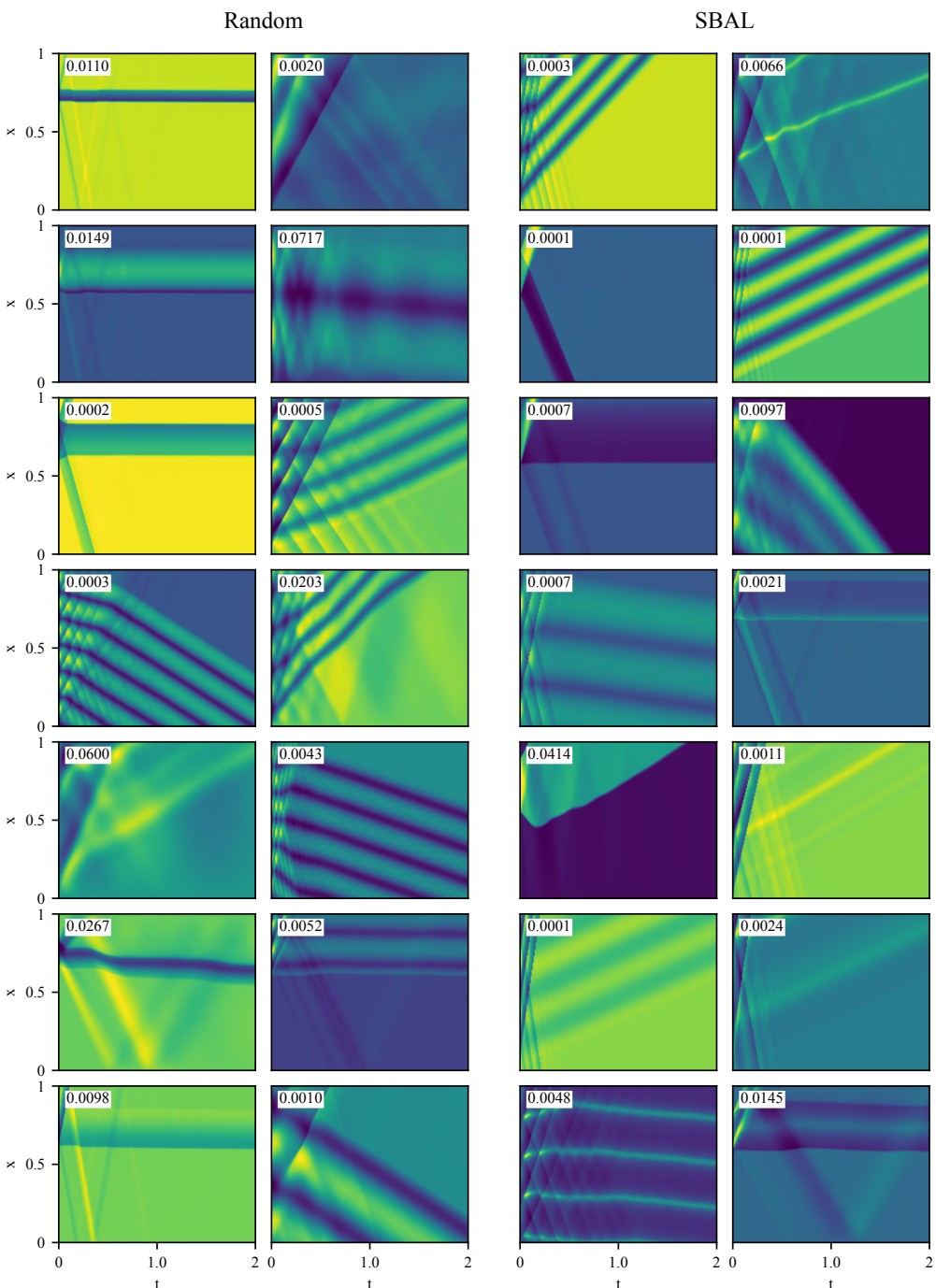

Figure 14: Example ground truth trajectories of random and SBAL on 1D CNS. The number on the top left of the trajectories shows the PDE parameters ($\eta = \zeta$).

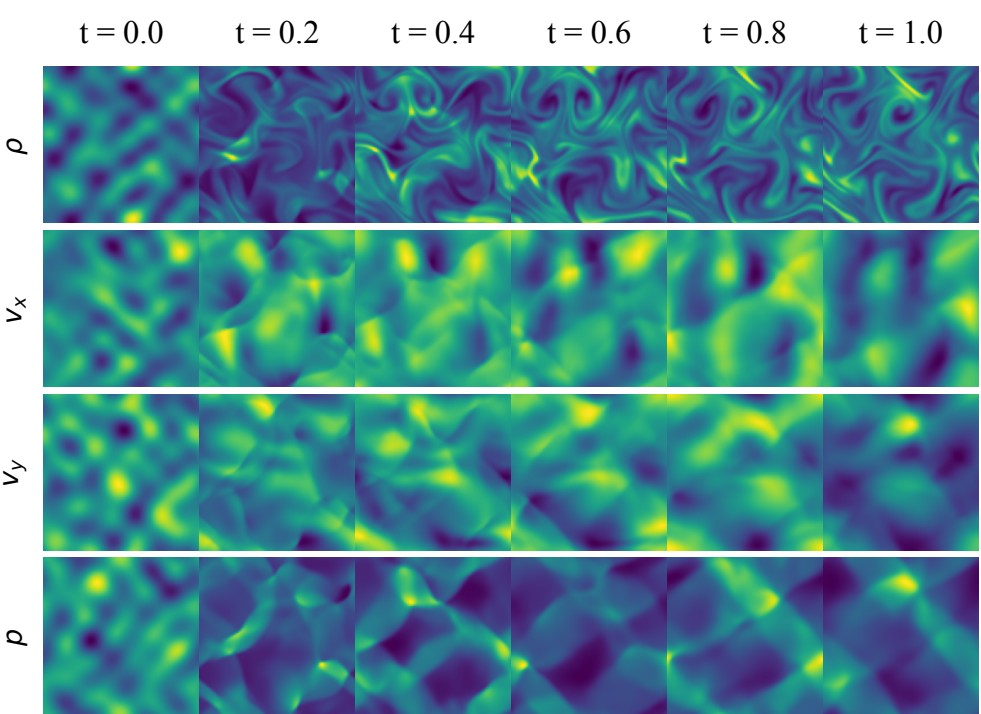

Figure 15: Example ground truth trajectory of 2D CNS.

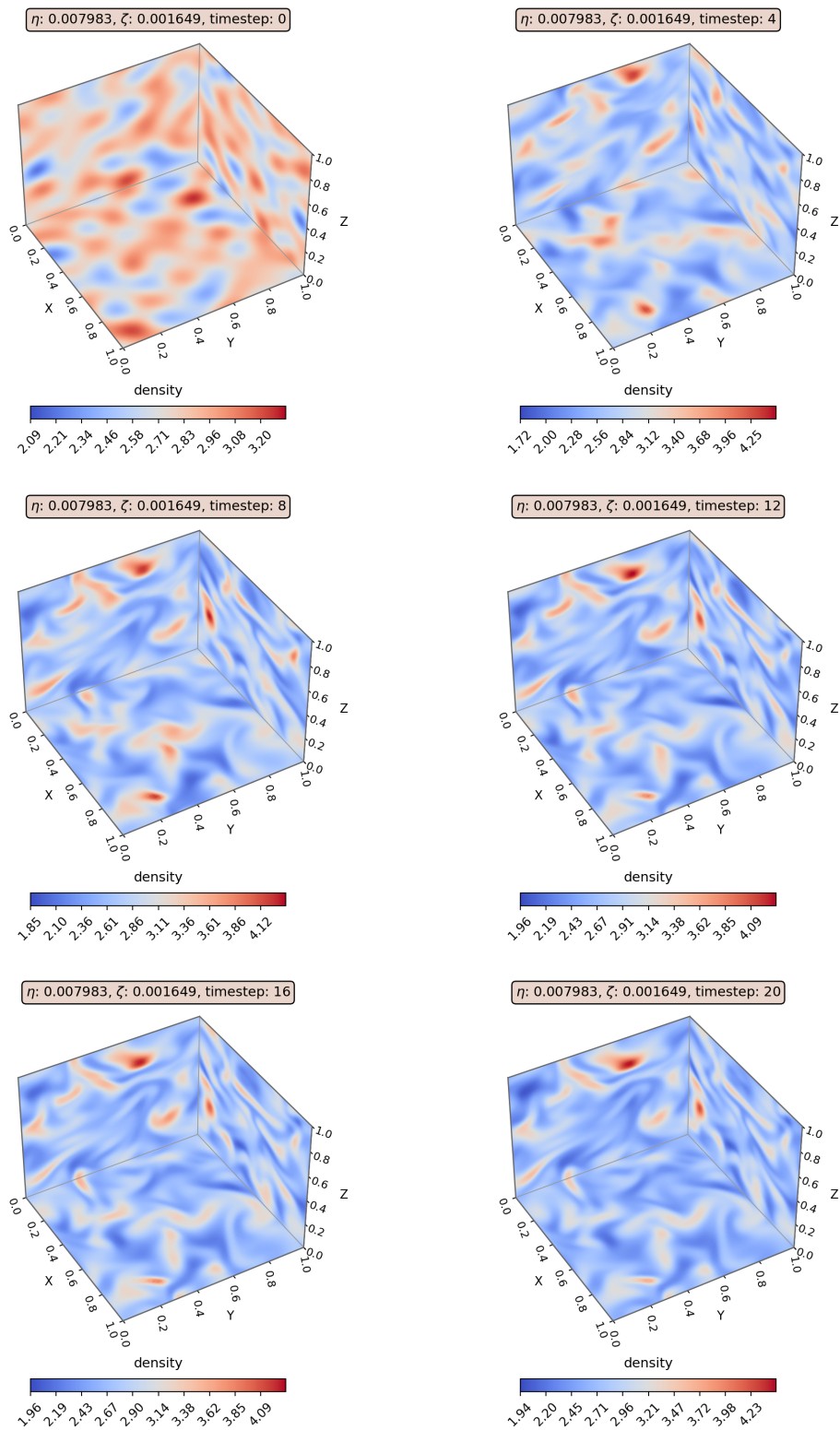

Figure 16: XY, YZ, and XZ planar views of the *density* channel from a random 3D CNS trajectory in the validation dataset representing a simulation of the compressible flow on the unit cube.

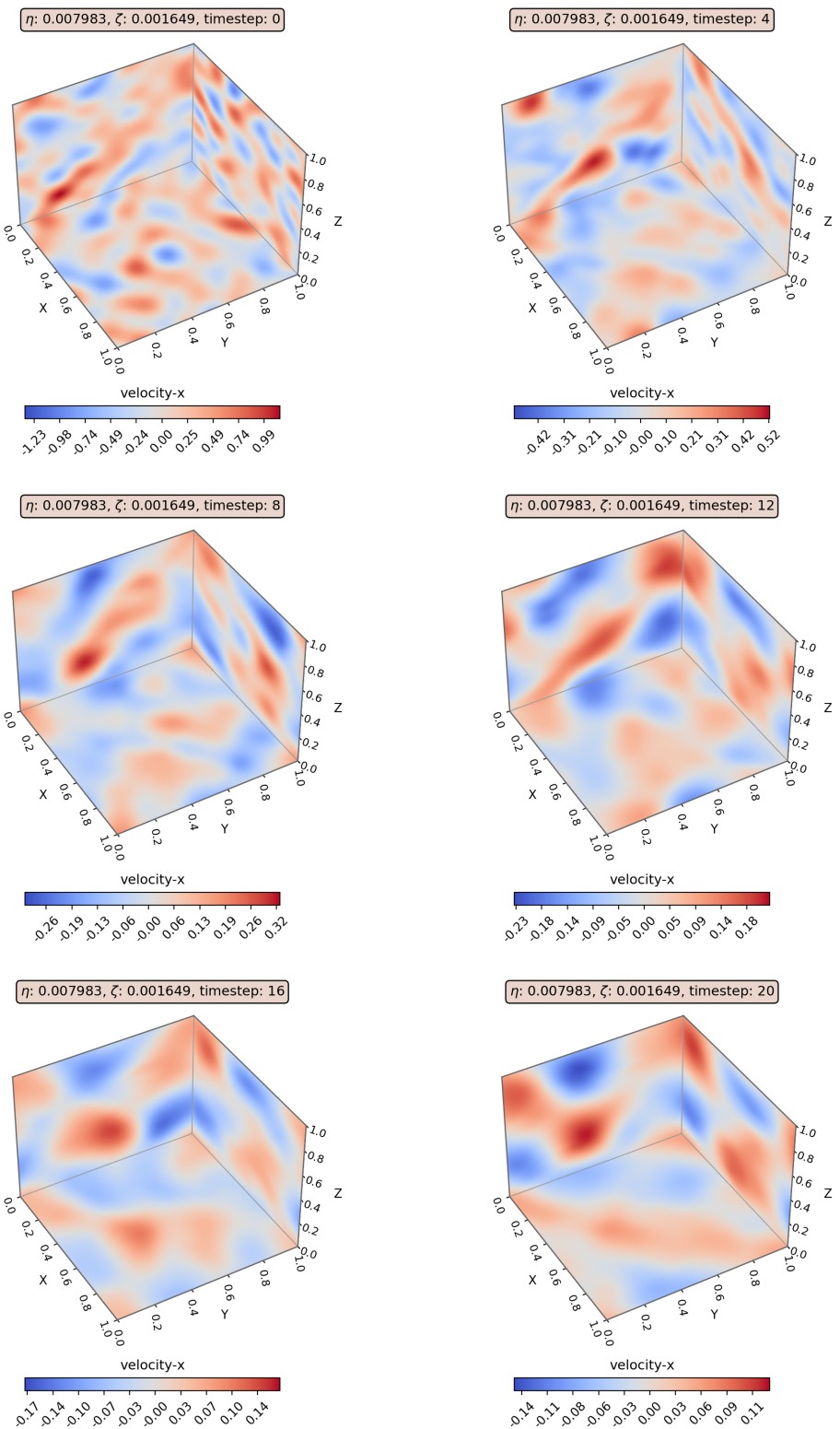

Figure 17: XY, YZ, and XZ planar views of the *velocity ($\vec{x}$)* channel from a random 3D CNS trajectory in the validation dataset showing a simulation of the compressible flow on the unit cube.

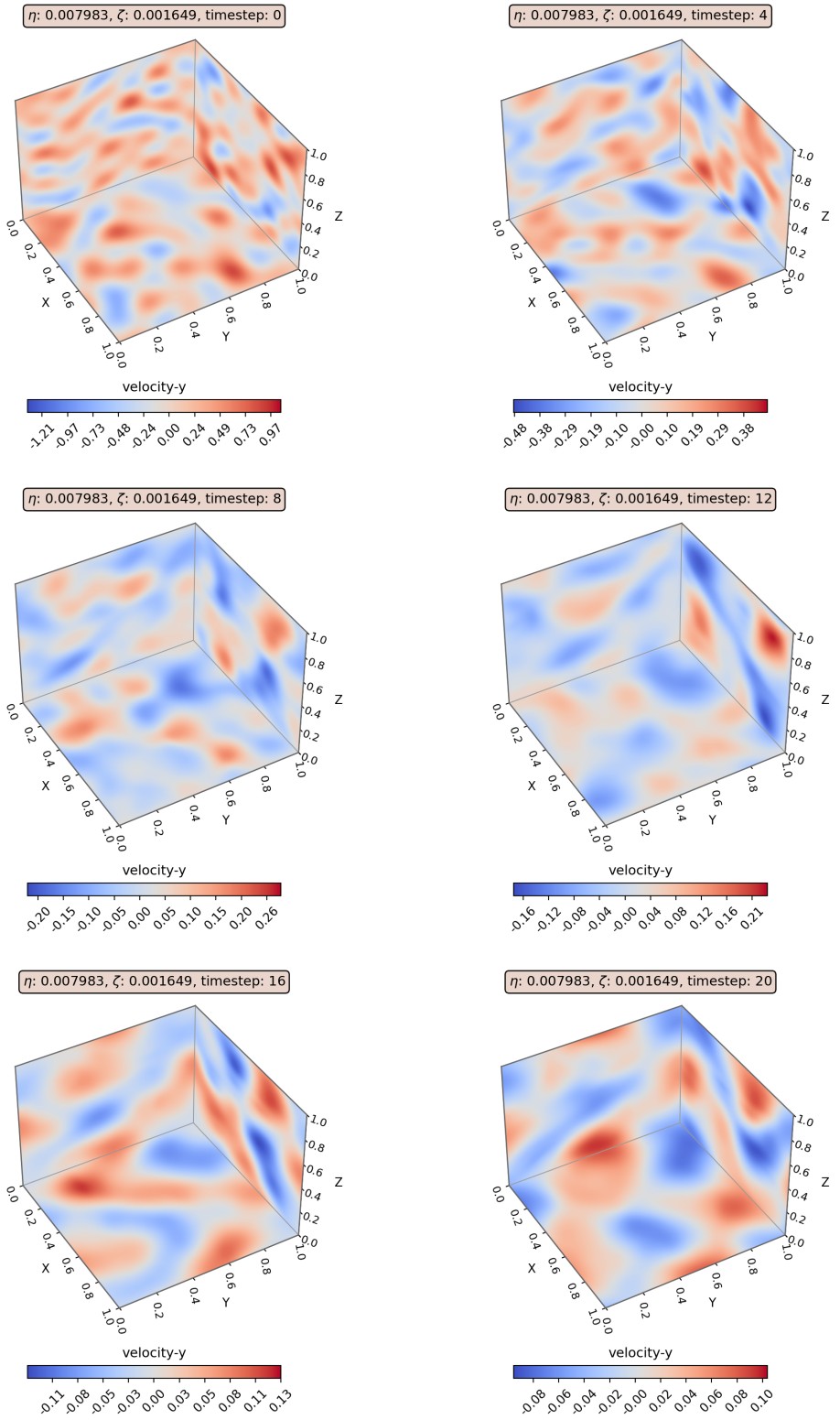

Figure 18: XY, YZ, and XZ planar views of the *velocity ($\vec{y}$)* channel from a random 3D CNS trajectory in the validation dataset showing a simulation of the compressible flow on the unit cube.

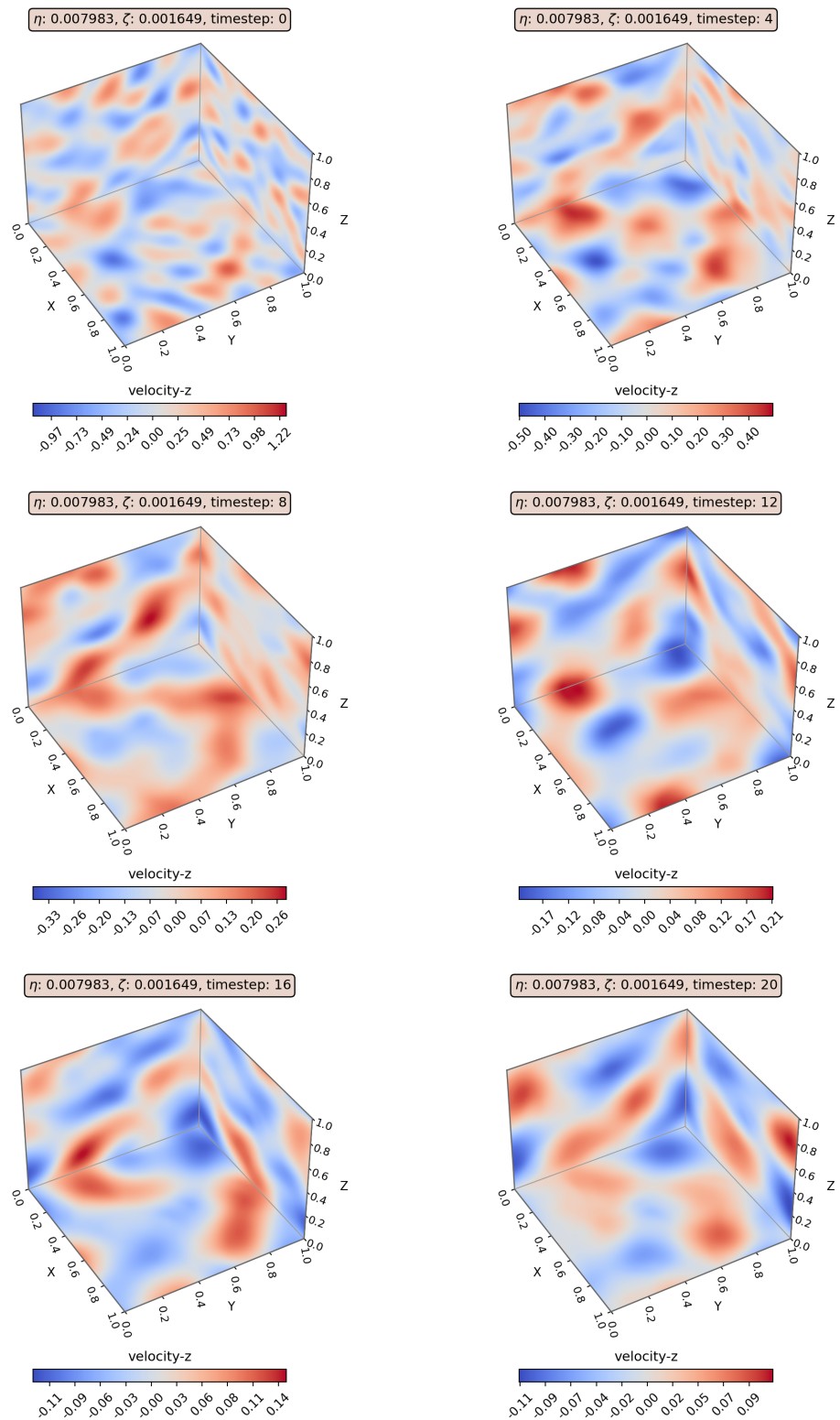

Figure 19: XY, YZ, and XZ planar views of the *velocity ($\vec{z}$)* channel from a random 3D CNS trajectory in the validation dataset showing a simulation of the compressible flow on the unit cube.

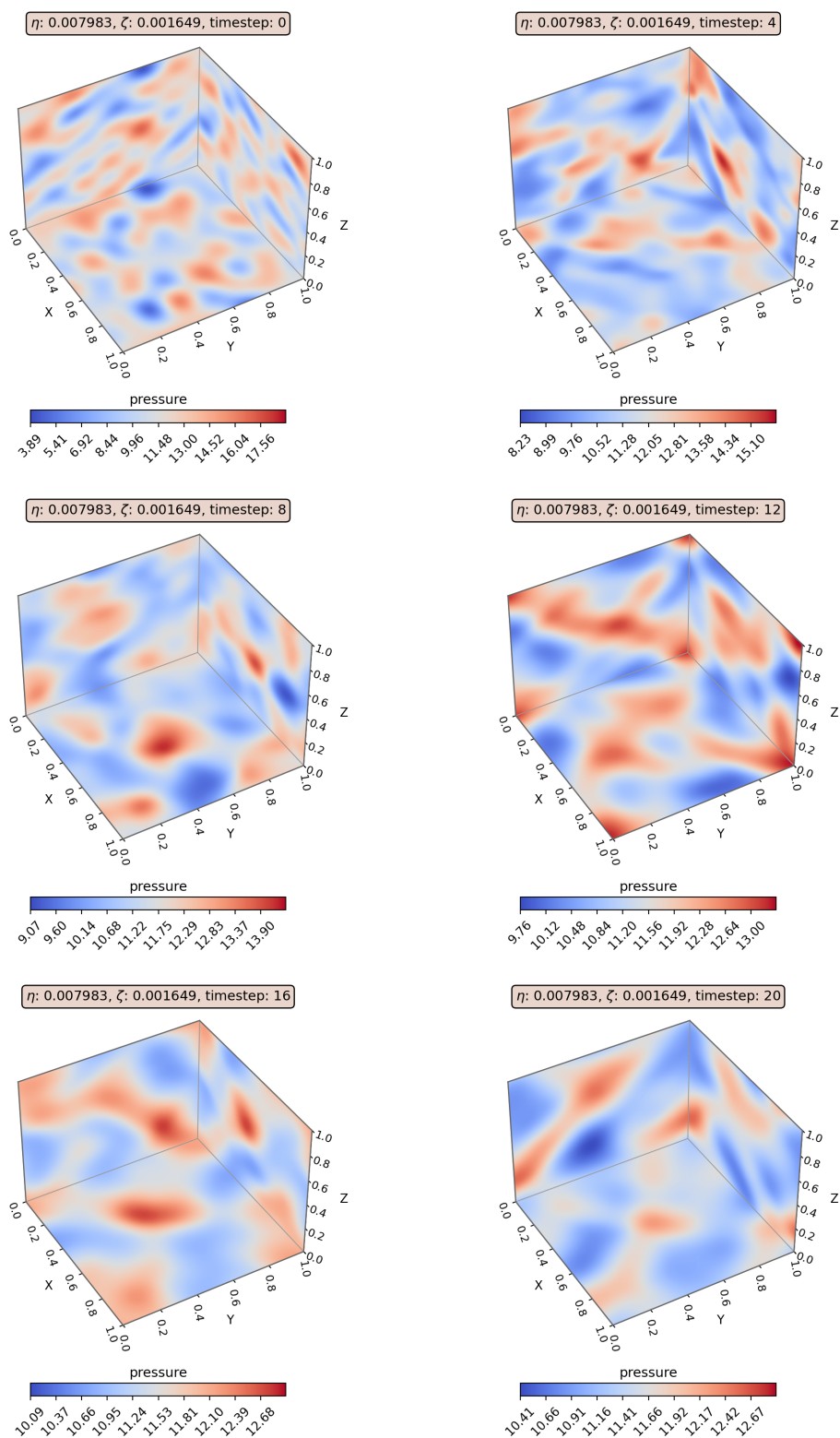

Figure 20: XY, YZ, and XZ planar views of the *pressure* channel from a random 3D CNS trajectory in the validation dataset representing a simulation of the compressible flow on the unit cube.

## G.2 IC PARAMETER MARGINAL DISTRIBUTIONS

Figures 21-26 show the marginal distributions of the random parameters of the IC generators, i.e., the random variables drawn which are then transformed using a deterministic function to the actual IC. For example, the KS IC generator draws amplitudes and phases from a uniform distribution and uses them afterward for the superposition of sine waves. If multiple numbers are drawn from each type of variable, we put them together, e.g., in the case of KS, multiple amplitudes are drawn for the different waves, but Figure 22 only shows the distribution of all amplitude variables mixed. The distribution curves for continuous variables are computed using kernel density estimation. The shaded areas (vertical lines for discrete variables) show the standard deviation between the marginal distributions of different random seeds.

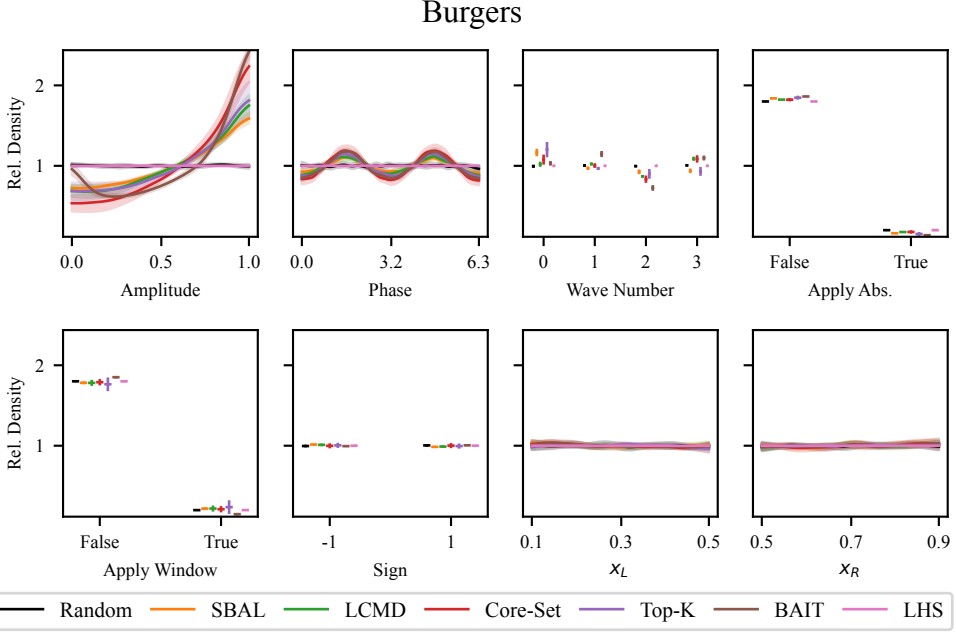

Figure 21: Marginal distribution of the parameters of the ICs sampled by the AL methods for Burgers. Displayed as the ratio to the density of the uniform distribution.

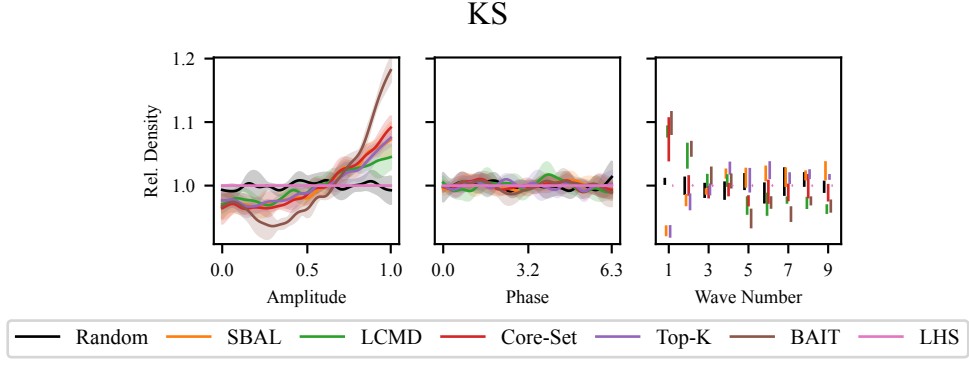

Figure 22: Marginal distribution of the parameters of the ICs sampled by the AL methods for KS. Displayed as the ratio to the density of the uniform distribution.

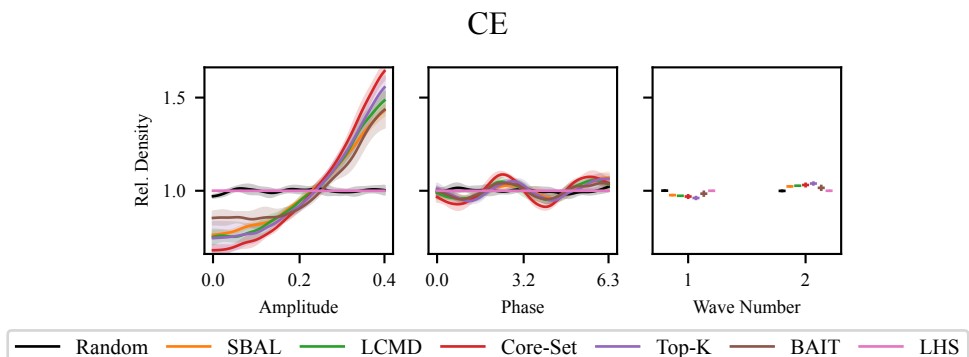

Figure 23: Marginal distribution of the parameters of the ICs sampled by the AL methods for CE. Displayed as the ratio to the density of the uniform distribution.

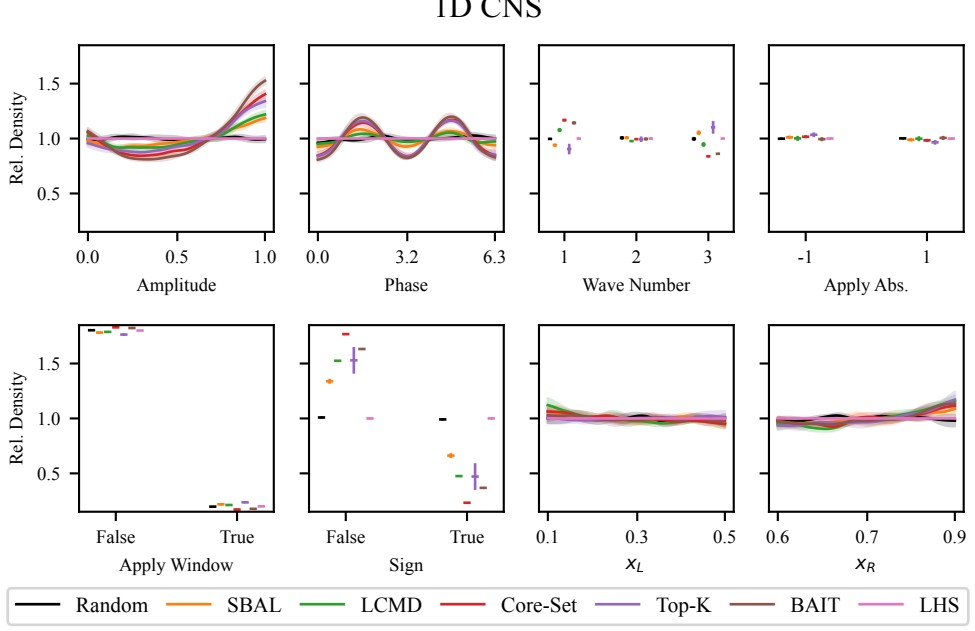

Figure 24: Marginal distribution of the parameters of the ICs sampled by the AL methods for 1D CNS. Displayed as the ratio to the density of the uniform distribution.

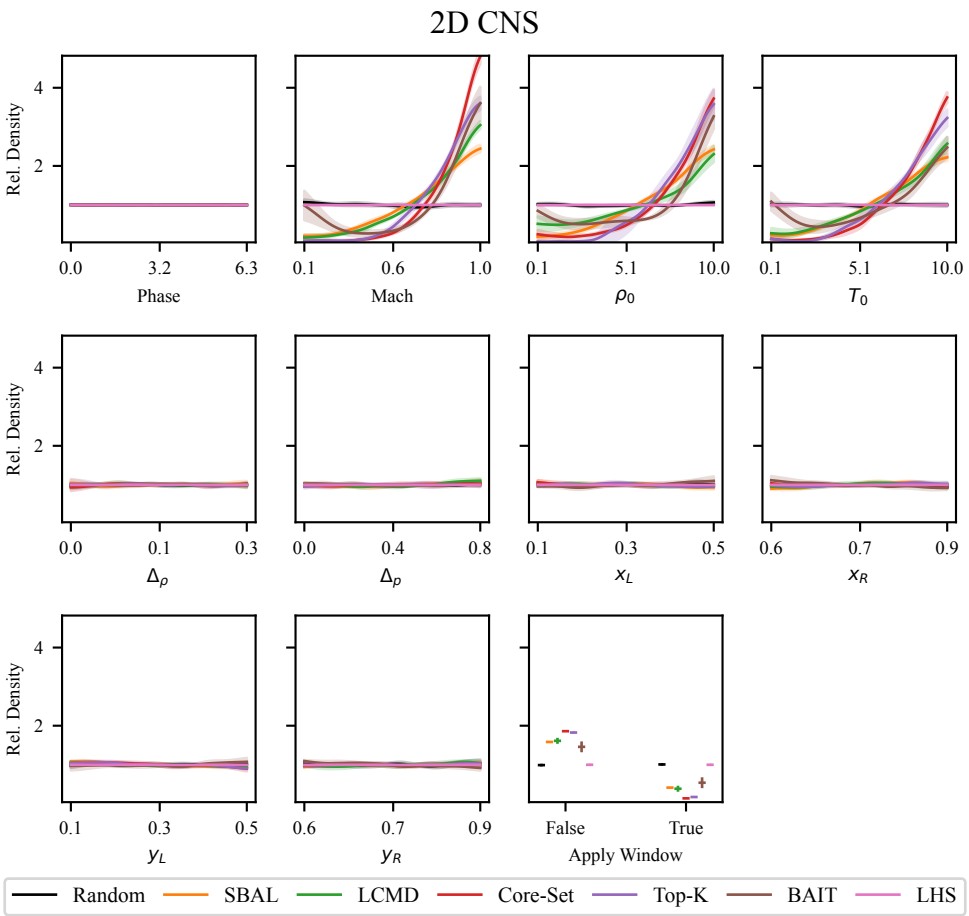

Figure 25: Marginal distribution of the parameters of the ICs sampled by the AL methods for 2D CNS. Displayed as the ratio to the density of the uniform distribution.

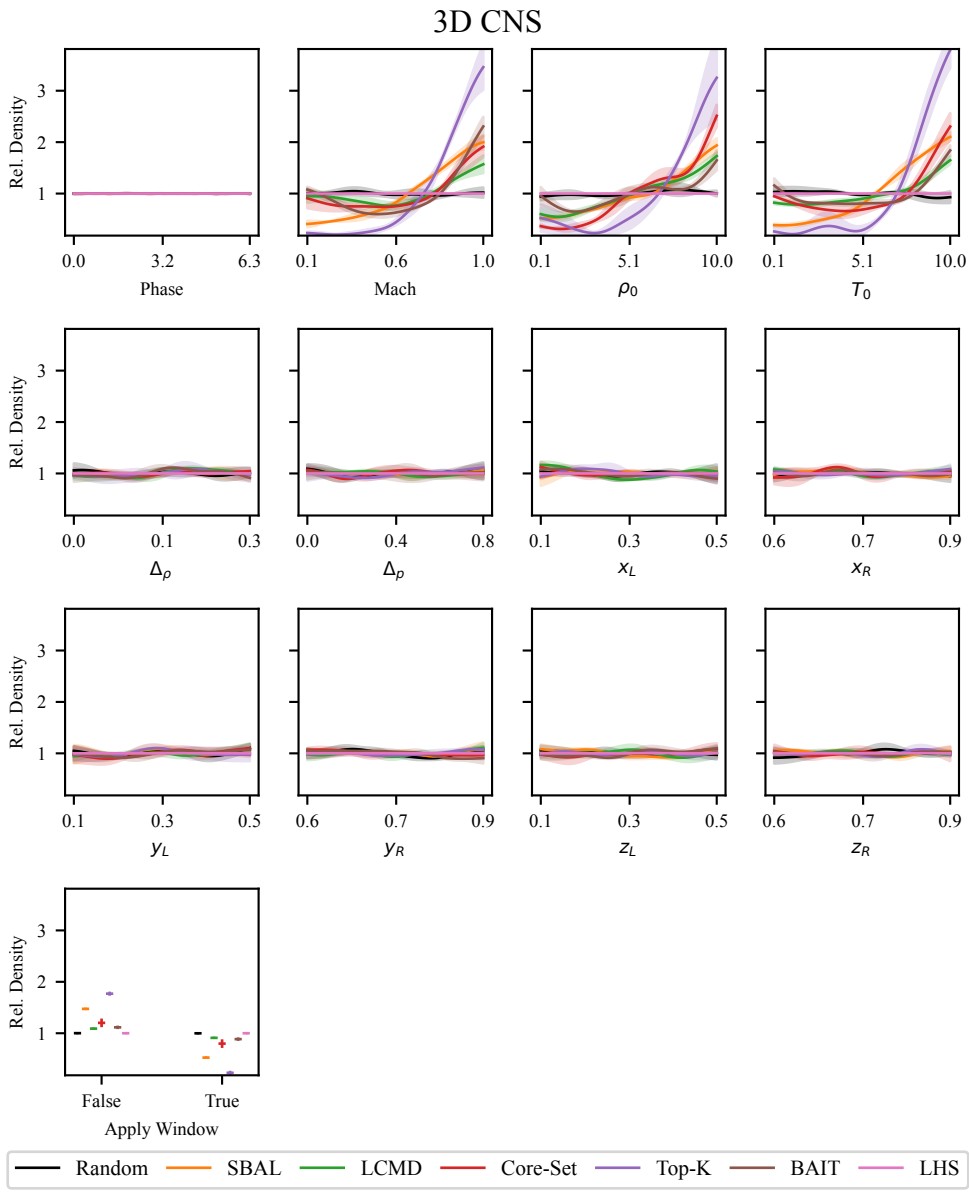

Figure 26: Marginal distribution of the parameters of the ICs sampled by the AL methods for 3D CNS. Displayed as the ratio to the density of the uniform distribution.

### G.3 PDE PARAMETER MARGINAL DISTRIBUTIONS

Similarly, Figures 27 and 28 show the KDE estimates of the dataset after the final AL iteration for the PDE parameters.

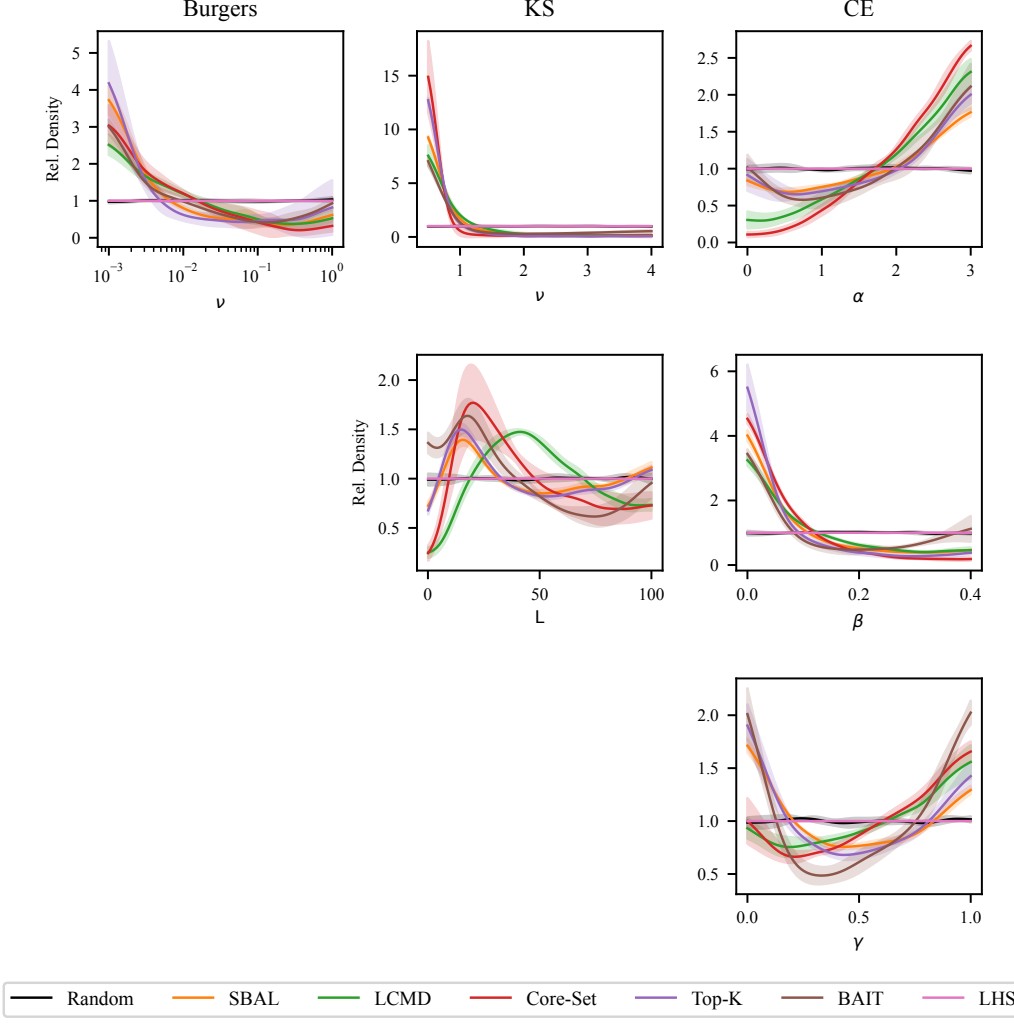

Figure 27: Marginal distribution of the PDE parameters of Burgers, KS, and CE, including the standard deviation between different runs. Displayed as the ratio to the density of the test distribution.

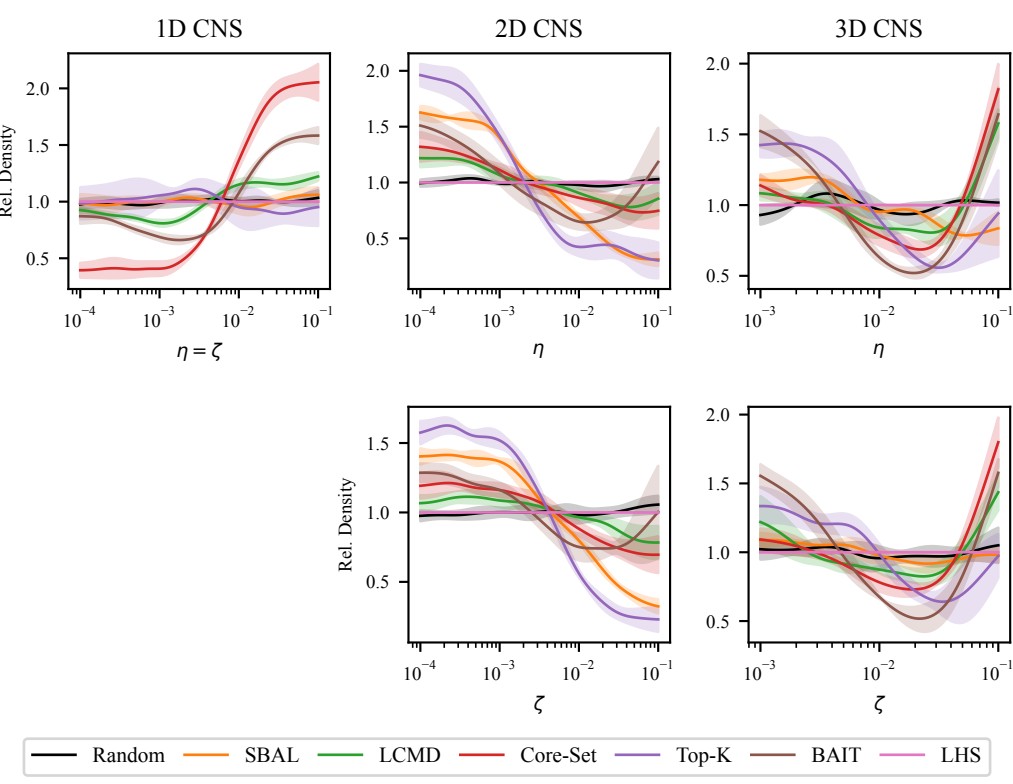

Figure 28: Marginal distributions of the 1D, 2D, and 3D CNS PDE parameters, including the standard deviation between different runs. Displayed as the ratio to the density of the test distribution.

