# OpenReview forum: "Active Learning for Neural PDE Solvers"
_ICLR.cc/2025/Conference — ICLR 2025 Poster_

### Official Review · Reviewer_KuN9 · 2024-11-04

**Soundness:** 3
**Presentation:** 3
**Contribution:** 2
**Rating:** 6
**Confidence:** 3

**Summary:**

The paper proposes a modular active learning (AL) framework for training surrogate models of partial differential equation (PDE) solvers. It introduces a numerical solver for generating PDE samples, several surrogate models, batch selection strategies, and acquisition functions designed for active learning within this framework.

**Strengths:**

The well-designed modular framework provides a solid foundation for further research on active learning in the context of PDEs.

**Weaknesses:**

1. Although the framework is tailored for PDE problems, the implemented acquisition functions are orthogonal to PDE problems i.e. they are AL methods that are used in general domains. As a framework of AL for PDE, at least some PDE-specific AL methods such as adaptive sampling [1], also mentioned in Related Work, should be also implemented.
1. The paper’s scope in terms of the surrogate models, acquisition functions, and types of PDEs studied is quite limited, impacting its practical applicability.

[1] W. Gao and C. Wang, Active Learning Based Sampling For High-dimensional Nonlinear Partial Differential Equations, Journal of Computational Physics, Vol. 475, 2023.

**Questions:**

No question

---

> ### Author Response · Authors · 2024-11-22
>
> We would like to thank the reviewer for the helpful feedback and for recognizing the modular framework we designed.
>
> >**W1:** Although the framework is tailored for PDE problems, the implemented acquisition functions are orthogonal to PDE problems i.e. they are AL methods that are used in general domains. As a framework of AL for PDE, at least some PDE-specific AL methods such as adaptive sampling [1], also mentioned in Related Work, should be also implemented.
>
> We would like to point out that the adaptive sampling methods presented in Gao and Wang [1] are used to select collocation points for PINNs, i.e., they select points in the spatiotemporal domain $(x, t)$, which are used to evaluate the PDE loss used in PINNs. In our work, we select the initial conditions and PDE parameters for which a numerical simulator generates the complete, discretized solution trajectory $u$. Using the PINN loss as a selection criterion, as in Gao and Wang [1], presupposes the use of the true loss of the model at that data point, which is not available in our setting and in active learning in general.
>
> The surrogate models considered in our AL4PDE benchmark are not trained with the PINN loss and cannot be trained in this manner since they do not model the solution as a function of $x$ and $t$ but instead output the complete solution at all grid points at once for the next time step.
>
> As an adaptation to our neural operator setting, we provide an experiment that uses the PINO [2] loss as a substitute for the uncertainty for SBAL (see Fig. 8b). It performs similarly to random selection.
>
> >**W2:** The paper’s scope in terms of the surrogate models, acquisition functions, and types of PDEs studied is quite limited, impacting its practical applicability.
>
> We would like to emphasize that the introduced AL methods and surrogate models can be flexibly replaced with any other depending on the user's target domain. Moreover, our new framework would reduce a practitioner's time to introduce SOTA AL approaches from scratch, which could be a significant contribution to the community.
>
> We have also expanded the scope by adding the following:
> - A new 1D PDE with non-periodic boundary conditions  (Fig. 4)
> - An experiment with 3D PDE (Fig. 4)
> - An experiment  with  Latin Hypercube sampling  (Fig. 4)
> - An adaption of BAIT [3] as an additional baseline  (Fig. 4)
> - Include the aforementioned PINO selection criterion (Fig. 8b)
>
> -----------------------
>
> [1] W. Gao and C. Wang, Active Learning Based Sampling For High-dimensional Nonlinear Partial Differential Equations, Journal of Computational Physics, Vol. 475, 2023.
> [2] Li, Zongyi, et al. "Physics-informed neural operator for learning partial differential equations." ACM/JMS Journal of Data Science 1.3 (2024): 1-27.
> [3] Ash, Jordan, et al. "Gone fishing: Neural active learning with fisher embeddings." Advances in Neural Information Processing Systems 34 (2021): 8927-8939.

---

> > ### Comment · Reviewer_KuN9 · 2024-11-25
> >
> > Thank you for clarifying the differences from adaptive sampling. I understand that the PDE-specific AL approach may have its limitations. The reason I raised the requirements for PDE-specific AL is that the contribution of your work appears to primarily lie in combining two independent domains (PDE learning and Active Learning). *However, such a combination could potentially be achieved using well-established Python repositories for PDE learning (e.g., [PDEBench](https://github.com/pdebench/PDEBench)) and active learning (e.g., [modAL](https://github.com/modAL-python/modAL), [libact](https://github.com/ntucllab/libact?tab=readme-ov-file), [scikit-activeml](https://github.com/scikit-activeml/scikit-activeml), and [regAL](https://git.rz.tu-bs.de/proppe-group/active-learning/regAL)).* Could you elaborate on what specifically differentiates your approach from simply integrating two independent frameworks? Based on my understanding of the paper, I feel its contribution (novelty) is somewhat limited.

---

> > > ### Author Response · Authors · 2024-11-25
> > >
> > > Thank you for your response. We would like to point out that our work makes several novel contributions:
> > >
> > > - We contribute an extensible benchmark codebase that integrates solvers, IC generators, and neural operators that take PDE parameters as input. This open-source framework will hopefully lead to additional AL innovations for PDE solvers. **Please note that the call for papers for ICLR specifically mentions benchmark papers, and the paper is placed in that category**.
> > > - We adapt existing batch active learning methods to the time-dependent parametric PDE setting. This is nontrivial due to the high-dimensional and spatio-temporal nature of the problems. For example, the setting breaks the assumption of a last-layer linear model made by many approaches, such as BAIT [1], since the autoregressive rollout applies the model (and hence the last layer) multiple times consecutively. Aggregating uncertainty or similarity information from these multiple layers is, therefore, challenging.
> > > - For the first time, we designed and conducted comprehensive experiments and reported their results on the behavior of batch active learning when applied to neural autoregressive PDE solvers. We have conducted numerous experiments showing the effects of different algorithmic choices and the consistent parameter distributions and, therefore, the reusability of the AL-generated datasets. Moreover, the experiments had to be designed carefully to provide a fair comparison of the AL methods to the random baseline. These experiments and empirical results are novel and required before new AL methods can be proposed and evaluated.
> > >
> > > -----------------
> > >
> > > [1] Ash, Jordan, et al. "Gone fishing: Neural active learning with fisher embeddings." Advances in Neural Information Processing Systems 34 (2021): 8927-8939.

---

> > > > ### Comment · Reviewer_KuN9 · 2024-11-26
> > > >
> > > > Thank you for the explanation. I finally understood that your work is not simply combining PDE learning and active learning. I hope your work will be helpful for many further works regarding PDE active learning. I will raise my score.

---

### Official Review · Reviewer_U4Je · 2024-11-04

**Soundness:** 3
**Presentation:** 4
**Contribution:** 3
**Rating:** 8
**Confidence:** 4

**Summary:**

Thie article introduces an active learning benchmark for neural PDE solver. It compare exploration-exploitation tradeoffs based uncertainty (epsitemic uncertainty of an ensemble of models with top-K and SBAL) or features (using dimensionality reduction using Gaussian sketching with Core-Set and LCMD). The authors then show a benchmark of these method on 1D and 2D parametric PDEs adding the baseline of sampling uniformly at random to represent the lack of active learning.

**Strengths:**

Contributing a benchmark in active learning for PDE solvers fills a needed gap in computational infrastructure for PDE solvers that is key to the central challenge of data efficiency.
The article is pedagogical and presents clearly the capability of the benchmark.

**Weaknesses:**

The authors should help compare methods of Bayesian active learning and those of the field of design of experiments (DoE), which is missing in the literature review. For example, instead of using a baseline of uniform sample, Latin Hypercube sampling should be provided, as well as more sophisticated DoE methods. This benchmark effort is an opportunity to bridge these areas of research and communities that try to solve the same problem with a slightly different point of view and a different approach.

The benchmark should broaden the UQ methods by connecting to existing efforts (for example, the open-source UQ 360). Any UQ method that provide a confidence interval should suffice for active learning as the spread of the confidence interval can be a proxy of the uncertainty.

In the implementation details, the tradeoffs of the choice of taking the spatial average over the features to make make feature-based AL translation invariant are not discussed. It seems that the averaging creates a significant dimensionality reduction that may outweigh the benefits of a translational invariance in terms of data efficiency.

All the implementations use periodic boundary conditions, which significantly limits the scope of applications.

**Questions:**

Could you please add baselines form DoE?

Could you link your benchmark code to existing open source code for uncertainty quantification (e.g. UQ 360)?

Could you explain the tradeoffs of reducing the dimensionality of features vs. implementing translational invariance in terms of data efficiency of the feature-based AL?

Could you add examples that do not use periodic boundary conditions?

---

> ### Author Response · Authors · 2024-11-22
>
> We thank the reviewer for the insightful questions and helpful suggestions and for acknowledging the importance of our active learning for the neural PDE solvers benchmark.
>
> > **W1:** The authors should help compare methods of Bayesian active learning and those of the field of design of experiments (DoE), which is missing in the literature review. For example, instead of using a baseline of uniform sample, Latin Hypercube sampling should be provided, as well as more sophisticated DoE methods. This benchmark effort is an opportunity to bridge these areas of research and communities that try to solve the same problem with a slightly different point of view and a different approach.
> > **Q1:** Could you please add baselines form DoE?
>
> Thank you for pointing out the additional references.  We added a short discussion on DoE methods applied to neural PDE solvers in the related work section,  Lines 152-157, and a more extended version in Appendix A.1, Lines 934-948. We also included an experiment with Latin Hypercube sampling, which performs similarly to random sampling (see Fig 4, top right). Additionally, we have added an adapted version of BAIT [1] (with the same averaged features we used as an input for LCMD), a Bayesian-motivated method for neural networks. BAIT performs similarly to LCMD on the CE PDE experiment (see Fig 4, top right). Many other DoE or Bayesian methods are, unfortunately, not directly applicable to our AL4PDE benchmark because we use neural networks that have high-dimensional output spaces and that are iterated for multiple time steps during inference (at least not without crude approximations as for our new experiment with BAIT, i.e., ignoring the autoregressive nature of the prediction). The autoregressive rollout breaks the last-layer linear model assumption typically made in these approaches [1].
>
> >**W2:** The benchmark should broaden the UQ methods by connecting to existing efforts (for example, the open-source UQ 360). Any UQ method that provide a confidence interval should suffice for active learning as the spread of the confidence interval can be a proxy of the uncertainty.
> >**Q2:** Could you link your benchmark code to existing open source code for uncertainty quantification (e.g. UQ 360)?
>
> Including more uncertainty quantification methods in the framework is an excellent idea. Similar to prior Bayesian methods, many uncertainty quantification methods are challenging to apply to the autoregressive, high-dimensional rollouts. The implementations in UQ 360 seem to target tabular ML problems. Consequently, it is not straightforward to integrate this library into our framework. Hence, we consider this future work.
>
> >**W3:** In the implementation details, the tradeoffs of the choice of taking the spatial average over the features to make make feature-based AL translation invariant are not discussed. It seems that the averaging creates a significant dimensionality reduction that may outweigh the benefits of a translational invariance in terms of data efficiency.
> >**Q3:** Could you explain the tradeoffs of reducing the dimensionality of features vs. implementing translational invariance in terms of data efficiency of the feature-based AL?
>
> We agree that the spatial averaging may remove important information. However, in Fig. 8c) we showed that it works better empirically for the considered PDE since it considers the translation equivariance of the periodic boundary condition. Finding a more expressive but invariant representation for LCMD would be an exciting idea for future work.
>
> >**W4:** All the implementations use periodic boundary conditions, which significantly limits the scope of applications.
> >**Q4:** Could you add examples that do not use periodic boundary conditions?
>
> Considering your suggestion, we have added an experiment with 1D compressible Navier-Stokes with an outgoing boundary condition in the main results to make the benchmark more extensive (Fig. 4).
>
>
> -----------------
>
> [1] Ash, Jordan, et al. "Gone fishing: Neural active learning with fisher embeddings." Advances in Neural Information Processing Systems 34 (2021): 8927-8939.

---

> > ### Comment · Reviewer_U4Je · 2024-11-22
> >
> > The authors have responded to my concerns in a convincing manner. I have revised my rating to 8: accept, good paper.

---

> > > ### Author Response · Authors · 2024-11-25
> > >
> > > Thank you for your quick reply and engaging with our rebuttal! We're pleased to have addressed your questions and concerns.

---

### Official Review · Reviewer_4k62 · 2024-11-04

**Soundness:** 3
**Presentation:** 3
**Contribution:** 2
**Rating:** 6
**Confidence:** 3

**Summary:**

This paper proposes a benchmark framework for neural PDE solvers under active learning (AL) settings (AL4PDE). It provides a modular benchmark with various parametric PDEs and AL methods. The experimental results show that AL significantly reduces average and worst case errors compared to random sampling and yields reusable datasets across experiments.

**Strengths:**

The paper is well-presented and easy to follow.

The proposed framework is novel in extending neural PDE methods with active learning methods.

The benchmark includes various batch selection strategies and neural PDE solvers, covering recent and classical works.

**Weaknesses:**

One key benefit of AL is data efficiency, which is also stressed in the paper. It is important to show how much data reduction can be achieved with reasonable model performance.
Current experiment section only shows performance comparison of different active learning methods and lacks the "offline" performance, which is training the model with full dataset and evaluate its performance.

At line $88$, the author claims "We demonstrate that using AL can result in more accurate surrogate models trained in less time." As mentioned above, this claim is not supported with empirical evidence as the experimental section only compares active learning performance, which cannot demonstrate improvement in accuracies regarding offline performance.

The novelty of this framework seems limited, as it is a combination of existing AL and neural PDE methods.

**Questions:**

Please refer to the strengths and weaknesses.

---

> ### Author Response · Authors · 2024-11-22
>
> We thank the reviewer for the helpful feedback and for noticing the novelty of our AL4PDE benchmark.
>
> > **W1:** One key benefit of AL is data efficiency, which is also stressed in the paper. It is important to show how much data reduction can be achieved with reasonable model performance. Current experiment section only shows performance comparison of different active learning methods and lacks the "offline" performance, which is training the model with full dataset and evaluate its performance.
>
> > **W2:** At line 88, the author claims "We demonstrate that using AL can result in more accurate surrogate models trained in less time." As mentioned above, this claim is not supported with empirical evidence as the experimental section only compares active learning performance, which cannot demonstrate improvement in accuracies regarding offline performance.
>
> We compare AL against the “Random” baseline in all experiments, which is equivalent to the “offline” performance in our setting. In contrast to the typical use of AL, where we have a fixed unlabeled dataset, there is no such dataset when training neural PDE solvers. Here, one has to generate new training trajectories with the (potentially computationally expensive) numerical PDE solvers. This is typically done on random initial conditions and PDE parameters. Thus, the AL methods are always compared to the random baseline for the same number of training trajectories in our main experiments, which does not use AL and samples the input uninformedly. Since the pool size is set to a large number (100,000 samples), it is computationally infeasible to compare the AL methods with a model trained with the fully labeled pool. However, we have added an experiment with a smaller total pool size in Appendix E, Fig. 10,  where the entire pool is used at the end of the experiment.
>
> >**W3:** The novelty of this framework seems limited, as it is a combination of existing AL and neural PDE methods.
>
> We would like to point out that our work makes several novel contributions:
> 1. We contribute an extensible benchmark codebase as an open-source framework to the research community that will hopefully lead to additional innovations in AL for PDE solvers. Please note that the call for papers for ICLR specifically mentions benchmark papers.
> 2. It adapts existing batch active learning methods to the time-dependent parametric PDE setting. This is nontrivial due to the high-dimensional and spatio-temporal nature of the problems.
> 3. We provide, for the first time, comprehensive experiments and their results on the behavior of batch active learning when applied to neural autoregressive PDE solvers. We have conducted numerous ablation experiments showing the effects of different algorithmic choices. The experiments had to be designed carefully to make a fair and in-depth comparison of the AL methods to the random baseline.

---

> ### Comment · Reviewer_4k62 · 2024-11-25
>
> Thank you for the detailed response and revisions. I am still a bit concerned regarding the offline setting. One possible setup would be using a smaller, completely labeled pool (i.e. 8192 inputs) to train a model without AL as the offline performance baseline, and with the same dataset pool, AL methods can be tested to see how different models / AL techniques can reach the offline performance with less data. Considering the time being, it would be great to add this in future revisions.
>
> Other than the offline setting, the authors have addressed my concerns well. I have raised the score.

---

> > ### Author Response · Authors · 2024-12-02
> >
> > Dear reviewer 4k62,
> >
> > Thank you for your response and constructive feedback. As far as we understand your suggestion, we have already done this in Appendix E, Figure 10, where the "random" line for tick 8192 (x-axis) corresponds to the RMSE (y-axis) of using all 8192 data points in the pool of size 8192 for training. Hence, in this figure, one can see when the AL methods reach the RMSE of the method trained with all pool data points. This is (considering confidence intervals) for about half of the data points for the AL method with a smaller pool size of 8192, and for about a quarter of the data size for the AL method with a larger pool size. We will clarify this in the final version of the paper.
> >
> > In addition, we will also add the same plot but where we use time on the x-axis to capture the additional time needed for the AL method with the larger pool size.
> >
> > Finally, we politely suggest that the reviewer might want to consider raising their score to an "accept" since we have addressed all shortcomings mentioned in the initial review. Thank you again for your helpful review and feedback either way.

---

### Official Review · Reviewer_zzgS · 2024-11-06

**Soundness:** 3
**Presentation:** 3
**Contribution:** 3
**Rating:** 8
**Confidence:** 3

**Summary:**

This paper provided a bechmark called AL4PDE, which unifies active learning (AL) with neural PDE solvers. Specifically, it studies how several state-of-the-art neural surrogate model may be applied to solve parametric PDEs under a solver-in-the-loop (AL) setting. A complete set of numerical experiments on various tasks is included to justify the effectiveness of AL based methods compared to methods based on random sampling.

**Strengths:**

Extensive numerical experiments on multiple PDEs are provided to validate the effectiveness of the proposed methodology. Also, details about the numerical experiments, such as the neural network models and training procedures, are included for the sake of completeness.

**Weaknesses:**

1. Though the authors have conducted a literature review on how AL has been used for solving other problems from scientific ML, such as PINN and direct prediction, it seems to the reviewer that the authors have missed a few important references like [1,2]. It might be meaningful for the authors to include these work and briefly discuss them in the introduction.

2. Given that this work aims for a complete benchmark on various tasks, the authors might consider including some more experiments on high-dimensional PDEs, just like the setting of [3].

**Questions:**

The parameter $c$ mentioned in line 96-97 (referred to as the field variables or channels) seems a bit ambiguous here as the following PDE doesn't contain anything about $c$. Would it be possible for the authors to provide more detailed explanation for the field variable/channel $c$ here? (This also highly relates to the parameter $N_c$ appearing in equations (3) and (5).)

References:

[1] Bruna, Joan, Benjamin Peherstorfer, and Eric Vanden-Eijnden. "Neural Galerkin schemes with active learning for high-dimensional evolution equations." Journal of Computational Physics 496 (2024): 112588.

[2] Gajjar, Aarshvi, Chinmay Hegde, and Christopher P. Musco. "Provable active learning of neural networks for parametric PDEs." In The Symbiosis of Deep Learning and Differential Equations II. 2022.

[3] Gao, Wenhan, and Chunmei Wang. "Active learning based sampling for high-dimensional nonlinear partial differential equations." Journal of Computational Physics 475 (2023): 111848.

---

> ### Author Response · Authors · 2024-11-22
>
> We thank the reviewer for the insightful questions and helpful suggestions and for appreciating the extensiveness.
>
> > **W1:** Though the authors have conducted a literature review on how AL has been used for solving other problems from scientific ML, such as PINN and direct prediction, it seems to the reviewer that the authors have missed a few important references like [1,2]. It might be meaningful for the authors to include these work and briefly discuss them in the introduction.
>
> Thank you for pointing out the additional sources. We have added the papers to the related work section in Line 141 and Line 152.
>
> > **W2:** Given that this work aims for a complete benchmark on various tasks, the authors might consider including some more experiments on high-dimensional PDEs, just like the setting of [3].
>
> We have added a 3D PDE experiment in the revised paper (see Figure. 4, Figures 16-20). The authors in [3], however, use elliptic, parabolic, and hyperbolic PDEs with up to 100 dimensions, which is computationally infeasible for Neural Operator type architectures since they require the solution to be discretized to a sufficient resolution and precomputed.
>
>
> >**Q1:** The parameter $c$
>  mentioned in line 96-97 (referred to as the field variables or channels) seems a bit ambiguous here as the following PDE doesn't contain anything about $c$
> . Would it be possible for the authors to provide more detailed explanation for the field variable/channel $c$
>  here? (This also highly relates to the parameter $N_c$
>  appearing in equations (3) and (5).)
>
> Thank you for pointing out the missing explanation. The field variables are used in the previous definition of the solution $u$ in Line 95. They refer to the different physical quantities that may be contained in a PDE, such as density, velocity, and pressure, in the case of the Navier-Stokes equation. Each of these quantities is included as a channel dimension in the discretized representation of the solution and needs to be predicted by the surrogate model. We have added an explanation in the revised paper in Line 105.

---

> > ### Comment · Reviewer_zzgS · 2024-11-22
> >
> > Thank you so much for your detailed rebuttal and revision of your manuscript. I'm satisfied with the authors' responses to the raised questions. Therefore, I'm raising my score.

---

> > > ### Author Response · Authors · 2024-11-25
> > >
> > > Thank you for your prompt response! We're glad our rebuttal & revisions have adequately addressed your questions and clarifications.

---

### Author Response · Authors · 2024-11-22
**Summary of Changes**

We thank all the reviewers for the insightful questions and the valuable feedback, which has helped us to improve the paper significantly. Considering your suggestions, we have uploaded a revised version with the following main additions (marked in blue on the paper):

1. An experiment with a non-periodic boundary condition (1D CNS) in Fig. 4
2. An experiment with a higher-dimensional PDE (3D CNS) in Fig. 4
3. An experiment with Latin Hypercube Sampling for the CE PDE in Fig. 4
4. An experiment with an adapted version of BAIT [1] as a Bayesian-inspired AL method for neural networks for the CE PDE in Fig. 4
5. An experiment that uses up the full pool size in Fig. 10
6. An experiment with a PINO [2] loss as a sampling criterion as an adaptation of adaptive sampling methods for PINNs in Fig 8b)
7. A discussion about  Design of Experiments (Sec. 2, Lines 152-157 and Appendix A.1, Lines 934-948)

[1] Ash, Jordan, et al. "Gone fishing: Neural active learning with fisher embeddings." Advances in Neural Information Processing Systems 34 (2021): 8927-8939.
[2] Li, Zongyi, et al. "Physics-informed neural operator for learning partial differential equations." ACM/JMS Journal of Data Science 1.3 (2024): 1-27.

---

### Meta-Review · Area_Chair_YeSj · 2024-12-19

**Metareview:**

This paper proposes a benchmark study for neural PDE solvers with active learning approaches. It provides a modular benchmark with various parametric PDEs as well as active learning methods. During the rebuttal, the authors have addressed the reviewers' questions and concerns on the completeness of experimental studies via revision of the paper and additional experiments. All the reviewers are positive about this paper after the rebuttal.

**Additional Comments On Reviewer Discussion:**

During the rebuttal, the authors have addressed the reviewers' questions and concerns on the completeness of experimental studies via revision of the paper and additional experiments.

---

### Decision · Program_Chairs · 2025-01-22

Accept (Poster)